# DISENTANGLEMENT AS IDENTIFIABLE PUSHFORWARD FACTORISATION

## ABSTRACT

We give a precise, general account of *disentanglement* for smooth generative models. For a decoder $g : \mathcal{Z} \to \mathcal{X}$ and factorised prior $p(z) = \prod_i p_i(z_i)$, we (i) define disentanglement as *factorisation of the pushforward density $p_\mu = g_\# p$ into one–dimensional "seam" factors (Def. D1); (ii) prove a canonical factorisation of $p_\mu$; and (iii) show that disentanglement is *equivalent* to two decoder conditions (C1–C2). Furthermore, under these conditions, the seam factors are *identifiable* up to permutation and sign. These results hold for general smooth pushforwards and are independent of VAEs. Specializing to Gaussian VAEs, we use an *exact* identity to show that diagonal posteriors (and $\beta$) promote C1–C2 in expectation, thereby explaining when and why VAEs exhibit disentanglement and how $\beta$ modulates it. Experiments illustrate this mechanism on Gaussian data, dSprites, and CelebA.

## 1 INTRODUCTION

A generative latent variable model is said to be *disentangled* when varying a single latent co-ordinate changes a single aspect of samples generated, e.g. object position or facial expression in an image. Variational Autoencoders (**VAE**s, Kingma & Welling, 2014; Rezende et al., 2014) and variants, (Higgins et al., 2017; Kim & Mnih, 2018; Chen et al., 2018), are often observed to disentangle data. This phenomenon is both of practical use, e.g. for controlled data generation, and intriguing as it is not knowingly designed into a VAE's training algorithm. Related phenomena are observed in samples from Generative Adversarial Networks (GANs) (Goodfellow et al., 2014) and diffusion models (Rombach et al., 2022; Pandey et al., 2022; Zhang et al., 2022; Yang et al., 2023).

While disentanglement lacks a formal definition, it is commonly associated with identifying ground truth *generative factors* of the data (Bengio et al., 2013). Thus a better understanding of disentanglement, and how it arises "for free" in a VAE, seems relevant to many areas of machine learning, its interpretability and potentially our understanding of the data; and may allow disentanglement to be induced reliably in domains where it cannot be readily perceived, e.g. gene sequence or protein modelling. We focus on disentanglement in VAEs, where it is well observed (Higgins et al., 2017; Burgess et al., 2018), with the expectation that a clearer understanding of it there, more as a property of the *data* than the model, may extend to other settings, such as state of the art diffusion models.

The cause of disentanglement in VAEs has been traced to *diagonal* posterior covariance matrices (Rolinek et al., 2019; Kumar & Poole, 2020), a common choice for computational efficiency. Approximate relationships suggest that diagonal covariances promote *orthogonality* between columns in a VAE decoder's Jacobian, a property empirically associated with disentangled features (Ramesh et al., 2018; Gresele et al., 2021). Taking inspiration from this, we develop a full theoretical explanation of disentanglement and how it arises in ($\beta$-)VAEs. Several core results are *model-agnostic* statements general to smooth pushforwards:

- a distributional definition of disentanglement in terms of independent factors (Definition 1, Fig. 1),
- a canonical factorisation of the pushforward density over the manifold (Lemma 5.1),
- an *iff* characterisation of disentanglement via decoder conditions (C1–C2) (Theorem 5.2), and
- proof that *independent factors are identifiable* up to natural symmetries (§6).

We then specialise to VAEs: using an *exact* Price/Bonnet identity, we show that Gaussian VAEs with *diagonal posteriors* (and $\beta$) *induce C1–C2 in expectation* over the encoder region (§4). This bridges prior empirical observations and VAE diagnostics with a distribution-level disentanglement and identifiability theory.

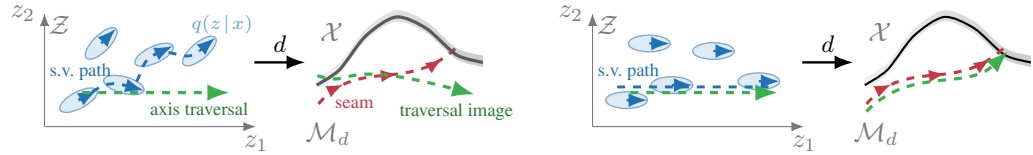

Figure 1: **Disentanglement: full vs diagonal posteriors ($\boldsymbol{\Sigma_x}$).** Right singular vectors $\boldsymbol{v}^i \in \mathcal{Z}$ (*blue*) of the decoder's Jacobian define *singular vector paths* (dashed blue); left singular vectors $\boldsymbol{u}^i$ define *seams* (dashed red). 1-D densities over seams factorise the manifold density. (*left*) with full posteriors, s.v. paths are not axis-aligned; the axis-traversal image in $\mathcal{X}$ (green) does *not* follow the seam. (*right*) under C1-C2 induced by diagonal posteriors, s.v. paths axis-align and the traversal image *follows the seam everywhere*, and 1-D densities over seams are independent, achieving disentanglement (D1).

## 2 BACKGROUND

**Notation**: Let $x \in \mathcal{X} \doteq \mathbb{R}^m$, $z \in \mathcal{Z} \doteq \mathbb{R}^d$ denote data and latent variables ($d \leq m$). For continuous $g : \mathcal{Z} \to \mathcal{X}$ differentiable at $z$, let $\boldsymbol{J}_z$ denote its Jacobian evaluated at $z$ ($[\boldsymbol{J}_z]_{ij} = \frac{\partial x_i}{\partial z_j}$) with singular value decomposition (SVD) $\boldsymbol{J}_z = \boldsymbol{U}_z \boldsymbol{S}_z \boldsymbol{V}_z^\top$ ($\boldsymbol{U}_z^\top \boldsymbol{U}_z = \boldsymbol{I}$, $\boldsymbol{V}_z^\top \boldsymbol{V}_z = \boldsymbol{V}_z \boldsymbol{V}_z^\top = \boldsymbol{I}$).[1] Let $s^i \doteq \boldsymbol{S}_{ii}$ denote the $i^{th}$ singular value, and $\boldsymbol{u}^i / \boldsymbol{v}^i$ the $i^{th}$ left/right singular vectors (columns of $\boldsymbol{U}/\boldsymbol{V}$). We consider continuous, injective functions $g$ differentiable *a.e.* (abbreviated ***c.i.d.a.e.***), which, e.g., admit ReLU networks. Such $g$ define a $d$-dimensional *manifold* $\mathcal{M}_g = \{g(z) \mid z \in \mathcal{Z}\}$ embedded in $\mathcal{X}$ (see Fig. 3). Since $g$ is injective, there exists a bijection between $\mathcal{Z}$ and $\mathcal{M}_g$; and $\boldsymbol{J}_z$ has full-rank, where defined.

**Latent Variable Model (LVM)**: We consider the generative model $p_\theta(x) = \int_z p_\theta(x|z) p(z)$ with independent $z_i$. For tractability, parameters $\theta$ are typically learned by maximising a lower bound (**ELBO**)

$$\int_x p(x) \log p_\theta(x) \;\; \geq \;\; \int_x p(x) \int_z q_\phi(z|x) \big( \log p_\theta(x|z) \; - \; \beta \log \tfrac{q_\phi(z|x)}{p(z)} \big) \, , \tag{1}$$

where $\beta = 1$ and $q_\phi(z|x)$ learns to approximate the model posterior, $q_\phi(z|x) \to p_\theta(z|x) \doteq \frac{p_\theta(x|z) p(z)}{p_\theta(x)}$.

**Variational Autoencoder (VAE)**: A VAE parameterises Eq. 1 with neural networks: a *decoder* network $d(z)$ parameterises the likelihood $p_\theta(x|z)$; and an *encoder* network parameterises the typically Gaussian posteriors $q_\phi(z|x) = \mathcal{N}(z; e(x), \Sigma_x)$ with *diagonal* $\Sigma_x$. The prior $p(z)$ is typically a standard Gaussian. We refer to a VAE with Gaussian likelihood $p_\theta(x|z) \doteq \mathcal{N}(x; d(z), \sigma^2 \boldsymbol{I})$ as a **Gaussian VAE** and to a Gaussian VAE with linear decoder $d(z) = \boldsymbol{D}z$, $\boldsymbol{D} \in \mathbb{R}^{m \times d}$ as a **linear VAE**.

**Disentanglement**: While not well defined, disentanglement typically refers to associating distinct semantically meaningful features of the data with distinct latent co-ordinates $z_i$, such that data generated by varying a single $z_i$ differ in a single semantic feature (Bengio et al., 2013; Higgins et al., 2017; Ramesh et al., 2018; Rolinek et al., 2019; Shu et al., 2019). While samples from a VAE exhibit disentanglement, setting $\beta > 1$ (a $\beta$-VAE) often enhances the effect, although at a cost to generative quality, e.g. blurrier images (Higgins et al., 2017; Burgess et al., 2018). Disentanglement relates closely to independent component analysis (**ICA**), which aims to recover statistically independent components of the data under the same LVM but with a deterministic observation model, $p_\theta(x|z) = \delta_{x-d(z)}$.

**Probabilistic PCA (PPCA)** (Tipping & Bishop, 1999) considers a linear Gaussian LVM

$$p(x|z) = \mathcal{N}(x; \boldsymbol{W}z, \sigma^2 \boldsymbol{I}) \qquad p(z) = \mathcal{N}(z; \boldsymbol{0}, \boldsymbol{I}) \tag{2}$$

where $\boldsymbol{W} \in \mathbb{R}^{m \times d}$ and $\sigma \in \mathbb{R}$.[2] The exact posterior $p(z|x)$ and MLE parameter $\boldsymbol{W}_*$ are fully tractable:

$$p(z|x) = \mathcal{N}(z; \tfrac{1}{\sigma^2} \boldsymbol{M} \boldsymbol{W}^\top x, \, \boldsymbol{M}), \;\; \boldsymbol{M} = (\boldsymbol{I} + \tfrac{1}{\sigma^2} \boldsymbol{W}^\top \boldsymbol{W})^{-1}; \quad \boldsymbol{W}_* = \boldsymbol{U_X} (\boldsymbol{\Lambda_X} - \sigma^2 \boldsymbol{I})^{1/2} \boldsymbol{R} \tag{3}$$

where $\boldsymbol{\Lambda_X} \in \mathbb{R}^{d \times d}$, $\boldsymbol{U_X} \in \mathbb{R}^{m \times d}$ contain the largest eigenvalues and respective eigenvectors of the covariance $\boldsymbol{X} \boldsymbol{X}^\top$; and $\boldsymbol{R} \in \mathbb{R}^{d \times d}$ is orthonormal ($\boldsymbol{R}^\top \boldsymbol{R} = \boldsymbol{I}$). As $\sigma^2 \to 0$, $\boldsymbol{W}_*$ approaches the SVD of the data matrix $\boldsymbol{X} = \boldsymbol{U_X} \boldsymbol{\Lambda_X}^{1/2} \boldsymbol{V_X}^\top \in \mathbb{R}^{m \times n}$, up to $\boldsymbol{V_X}$ (classical PCA). The model is *unidentified* since $\boldsymbol{R}$ is arbitrary, allowing uncountably infinite solutions. While $\boldsymbol{W}_*$ can be computed analytically, it can also be learned by maximising the ELBO (Eq. 1): letting $p_\theta(x|z) = \mathcal{N}(x; \boldsymbol{D}z, \sigma^2 \boldsymbol{I})$ and iteratively computing the optimal posterior (Eq. 3, left) and maximising w.r.t. $\boldsymbol{D}$ (we refer to this as **PPCA**$^{EM}$).

---

[1]To lighten notation, explicit dependence of $\boldsymbol{U}, \boldsymbol{V}, \boldsymbol{S}, \boldsymbol{u}^i, \boldsymbol{v}^i, s^i$ on $z$ is often suppressed where context is clear.
[2]We assume that data is centred, which is equivalent to including a mean parameter (Tipping & Bishop, 1999).

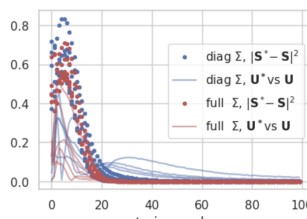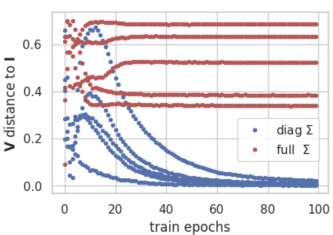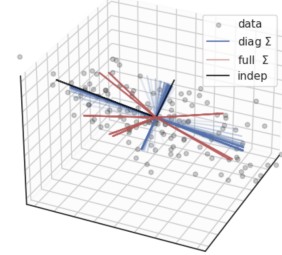

Figure 2: **An LVAE breaks rotational symmetry.** (*l*) both full-$\Sigma_x$ and diagonal-$\Sigma_x$ VAEs fit the data, i.e. learn ground truth parameters $\boldsymbol{U}_*$, $\boldsymbol{S}_*$ (all losses $\to 0$); (*c*) only in diagonal-$\Sigma_x$ VAEs do right singular vectors of the Jacobian, $\boldsymbol{v}^i \in \mathcal{Z}$, align with standard basis vectors $\boldsymbol{z}_i$, i.e. $\boldsymbol{V}_* \to \boldsymbol{I}$ (blue plots $\to 0$); (*r*) images of $\boldsymbol{z}_i$: full-$\Sigma_x$ VAEs map $\boldsymbol{z}_i$ to arbitrary directions (red), but diagonal-$\Sigma_x$ VAEs learn (later epochs darker) to map $\boldsymbol{z}_i$ to the data's independent components (black, i.e. blue $\to$ black).

**Linear VAE**: An LVAE assumes the same linear LVM as PPCA (Eq. 2) and models the likelihood $p_\theta(x|z) = \mathcal{N}(x; \boldsymbol{D}z, \sigma^2 \boldsymbol{I})$ as in PPCA$^{EM}$, differing only in *approximating* the posterior by $q_\phi(z|x) = \mathcal{N}(z; \boldsymbol{E}x, \Sigma)$, rather than computing the optimal $p_\theta(z|x)$. Surprisingly though, an LVAE with *diagonal* posterior covariances loses the rotational ambiguity of PPCA (Lucas et al., 2019), since

$$\Sigma_* \overset{(3(l))}{=} (\boldsymbol{I} + \tfrac{1}{\sigma^2} \boldsymbol{W}_*^\top \boldsymbol{W}_*)^{-1} \overset{(3(r))}{=} \sigma^2 \boldsymbol{R}^\top \boldsymbol{\Lambda}_{\boldsymbol{x}}^{-1} \boldsymbol{R}. \tag{4}$$

Thus for $\Sigma$ to be optimal *and* diagonal, $\boldsymbol{R}$ must belong to a finite set of signed permutations, hence the optimal decoder $\boldsymbol{D}_* = \boldsymbol{U}_{\boldsymbol{x}} (\boldsymbol{\Lambda}_{\boldsymbol{x}} - \sigma^2 \boldsymbol{I})^{1/2}$ is unique up to permutation/sign (see Fig. 2). We will see that this effect, due to diagonal posterior covariances, is in fact (linear) *disentanglement* (§3).

**Further notation**: Under the LVM above, we define a deterministic **generative function** $g : \mathcal{Z} \to \mathcal{X}$ as the map from latent variables to means $g(z) = \mathbb{E}[x|z]$, that lie on a manifold $\mathcal{M}_g = \{g(z)\} \subseteq \mathcal{X}$ (**mean manifold**) with push-forward density $p_\mu \doteq g_\# p_z$ (**manifold density**). We will focus on Gaussian VAEs, where the data density $p(x)$ is given by adding Gaussian noise to $p_\mu$, i.e. convolving it with a Gaussian kernel. It is known that such data densities match if and only if their manifold densities $p_\mu$ match (e.g. Khemakhem et al., 2020), hence we focus on the manifold density $p_\mu$.

## 3 DISENTANGLEMENT

We now define disentanglement; illustrate it for the linear case, justifying our disentanglement claim for LVAEs in §2; and work up to explaining how it arises in a (non-linear) Gaussian VAE. (See Fig. 1)

**Definition D 1** (**Disentanglement**). *Let* $g : \mathcal{Z} \to \mathcal{X}$ *be c.i.d.a.e.. We say* $p_\mu$ *is* disentangled *if, for each* $z \in \mathcal{Z}$, *there exist 1-D densities* $\{f_i\}$ *such that* $p_\mu$ factorises *as*

$$p_\mu\big(g(z)\big) = \prod_{i=1}^{d} f_i\big(u_i(z)\big), \tag{5}$$

*where each factor* $f_i$ *is the* 1-D push-forward *of* $p(z_i)$ *along the* axis-aligned *line obtained by moving in the* $i$-*th latent coordinate while keeping all others fixed;* $u_i$ *is the co-ordinate of* $g(z)$ *along the image of that line; and random variables* $\{u_i(z)\}$ *are mutually* independent *under* $z \sim p(z)$.

**D**1 defines disentanglement as *factorisation of the pushforward density* $p_\mu$ into 1-D factors, precisely the distributional independence expected when "changing one factor leaves the others unaffected."

**LVAE disentanglement**: Consider an LVAE with diagonal posterior covariance $\Sigma$ and decoder $d(z) = \boldsymbol{D}z$, $\boldsymbol{D} \in \mathbb{R}^{m \times d}$ (§2). The mean manifold $\mathcal{M}_d \doteq \{\mu = \boldsymbol{D}z \mid z \in \mathcal{Z}\}$ is linear with Gaussian density $p_\mu = \mathcal{N}(\mu; \boldsymbol{0}, \boldsymbol{D}\boldsymbol{D}^\top)$ (e.g. see Fig. 2, *right*), From §2, the SVD of the data matrix defines the optimal decoder $\boldsymbol{D}_* = \boldsymbol{U}_* \boldsymbol{S}_* \boldsymbol{V}_*$ (i.e. $\boldsymbol{U}_* \doteq \boldsymbol{U}_{\boldsymbol{x}}$, $\boldsymbol{S}_* \doteq (\boldsymbol{\Lambda}_{\boldsymbol{x}} - \sigma^2 \boldsymbol{I})^{1/2}$ and $\boldsymbol{V}_* = \boldsymbol{R} = \boldsymbol{I}$ due to diagonal $\Sigma$). As for any Gaussian, $p_\mu$ factorises as a product of independent 1-D Gaussians along eigenvectors of its covariance $\boldsymbol{D}_* \boldsymbol{D}_*^\top = \boldsymbol{U}_* \boldsymbol{S}_*^2 \boldsymbol{U}_*^\top$, i.e. columns $\boldsymbol{u}^i$ of $\boldsymbol{U}_*$, hence $p_\mu = \prod_i \mathcal{N}(u_i; 0, s^{i2})$ where $u_i \doteq \boldsymbol{u}^{i\top} \mu \in \mathbb{R}$. Since $u_i = \boldsymbol{u}^{i\top} \boldsymbol{D}_* z = s^i z_i$, each $u_i$ depends *only* on a distinct $z_i$, a co-ordinate in the standard basis of $\mathcal{Z}$ (over which densities are independent 1-D Gaussian). Thus:

- $p_\mu$ factorises as a product of independent push-forward densities $f_i(\mu) = \mathcal{N}(\boldsymbol{u}^{i\top} \mu; 0, s^{i2})$; and
- the decoder maps each axis-aligned direction $z_i$ to a distinct factor $f_i$,

satisfying D1. Note that synthetic data $\mu = \boldsymbol{D}z$ generated by re-sampling $z_i$, holding $z_{j \neq i}$ constant, differ only in component (or "feature") $u_i$, agreeing with the common perception of disentanglement.

**Dropping diagonality**: To emphasise that disentanglement depends on diagonal posteriors, we consider *full* posterior LVAEs, where $\boldsymbol{R} \neq \boldsymbol{I}$ in general. The above argument follows except that columns $\boldsymbol{r}^i$ of $\boldsymbol{R}$ in $\mathcal{Z}$ map to independent $\boldsymbol{u}^i$ directions in $\mathcal{X}$. Meanwhile, standard basis vectors in $\mathcal{Z}$ map in directions $\boldsymbol{u}^{i\top}\boldsymbol{R}$, which are arbitrary with respect to $\boldsymbol{u}^i$ directions. Hence axis-aligned traversals in latent space correspond to several *entangled* components $u_i$ changing in generated samples. We demonstrate this empirically in Fig. 2 (see caption for details).

## 4 FROM DIAGONAL POSTERIORS TO DECODER CONSTRAINTS

Prior works draw a link between disentanglement in Gaussian VAEs and diagonal posteriors from an *approximate* relationship between optimal posteriors and decoder derivatives (Rolinek et al., 2019; Kumar & Poole, 2020). In fact, this relationship is *exact* by the Price/Bonnet Theorem and Opper & Archambeau (2009): the ELBO with Gaussian posteriors is optimised when their covariances satisfy

$$\Sigma_x^{-1} \;=\; \boldsymbol{I} - \tfrac{1}{\beta}\mathbb{E}_{q(z|x)}[\mathsf{L}_z(x)] \;\stackrel{*}{=}\; \boldsymbol{I} + \tfrac{1}{\beta\sigma^2}\mathbb{E}_{q(z|x)}[\boldsymbol{J}_z^\top \boldsymbol{J}_z - (x - d(z))^\top \mathsf{H}_z]\,, \qquad (6)$$

where $\mathsf{L}_z(x) = \nabla_z^2 \log p_\theta(x|z)$ is the log likelihood Hessian; and $\boldsymbol{J}_z \doteq \frac{dx}{dz}$ and $\mathsf{H}_z \doteq \frac{d^2 x}{dz^2}$ are the Jacobian and Hessian of the decoder (all terms evaluated at $z \in \mathcal{Z}$). Step two (*) assumes the likelihood is Gaussian. Eq. 6 immediately generalises the classical linear result in Eq. 3 and relates $\sigma^2 \doteq \mathrm{Var}[x|z]$ and $\Sigma_x \doteq \mathrm{Var}[z|x]$, showing that (un)certainty in x and z go hand in hand, as expected.

Importantly to disentanglement, Eq. 6 shows that diagonal $\Sigma_x$ constrains derivatives of the decoder. In practice, the $\boldsymbol{J}_z^\top \boldsymbol{J}_z$ term *alone* is found to be approximately diagonal (Fig. 3, *left*) (Rolinek et al., 2019; Kumar & Poole, 2020), suggesting that *each* term diagonalises. We thus formalise this as a property and consider its implications (C1-C2), which will prove equivalent to disentanglement:

**Property P1.** $\boldsymbol{J}_z^\top \boldsymbol{J}_z$ and $(x - d(z))^\top \mathsf{H}_z$ in Eq. 6 are each diagonal for z concentrated around $\mathbb{E}[z|x]$.

**Lemma 4.1 (Disentanglement constraints).** *For a trained Gaussian VAE and* $x, z$ *satisfying P1:*

*C1) Right singular vectors* $\boldsymbol{V}_z$ *of the decoder Jacobian* $\boldsymbol{J}_z$ *are standard basis vectors, i.e. after relabeling/sign flips of the latent axes, we have* $\boldsymbol{V}_z = \boldsymbol{I}$;

*C2) The matrix of partial derivatives of singular values* $(\frac{\partial s_i}{\partial z_j})_{i,j}$ *is diagonal, i.e.* $\frac{\partial s_i}{\partial z_j} = 0$ *for all* $i \neq j$.

*Proof.* See Appendix A. **C1** follows from the SVD of $\boldsymbol{J}_z$; **C2** from observing that directions $r(z) = x - d(z) \in \mathcal{X}$ of the directed Hessian term are, to a first approximation, tangent to the manifold.

**Why** $\beta$ **affects disentanglement**: Setting $\beta > 1$ in Eq. 1 is found to enhance disentanglement (Higgins et al., 2017; Burgess et al., 2018). We show that $\beta$ implicitly controls the likelihood variance in Appendix B and thus $\beta > 1$ dilates posteriors (while the $\beta$-ELBO remains a valid objective). We will show that C1-C2 equate to disentanglement (§5), thus Eq. 6 suggests a rationale for why $\beta > 1$ enhances disentanglement: it broadens the regions (i.e. posteriors) over which decoder derivatives are diagonalised, hence where disentanglement constraints C1-C2 are encouraged; also increasing the overlap of posteriors where multiple constraints apply simultaneously (see Fig. 6, *right*).

## 5 FROM DECODER CONSTRAINTS TO DISENTANGLEMENT

To see how disentanglement relates to constraints C1-C2, defined in terms of the Jacobian SVD of the generative function $g$ (or decoder $d$), we consider the Jacobian SVD in detail.

**The Jacobian SVD**: For $\boldsymbol{J}_z = \boldsymbol{U}\boldsymbol{S}\boldsymbol{V}^\top$, singular vectors (columns of $\boldsymbol{V}$, $\boldsymbol{U}$) respectively define (local) orthonormal bases: the $\boldsymbol{V}$-basis, $\{\boldsymbol{v}^i\}$ for $\mathcal{Z}$ at $z$; and the $\boldsymbol{U}$-basis, $\{\boldsymbol{u}^i\}$ for the tangent space to $\mathcal{M}_g$ at $x = g(z)$. Letting $v \doteq \boldsymbol{V}^\top z$ and $u \doteq \boldsymbol{U}^\top x$ denote a point $z$ and its image $x = g(z)$ in those bases, the chain rule gives an interpretation of the Jacobian's SVD, $\boldsymbol{J}_z = \boldsymbol{U}\boldsymbol{S}\boldsymbol{V}^\top = \frac{\partial x}{\partial u}\frac{\partial u}{\partial v}\frac{\partial v}{\partial z}$: $\boldsymbol{U}$ and $\boldsymbol{V}^\top$ are simply local co-ordinate systems in each domain, and $\boldsymbol{S} = \frac{du}{dv}$ is the Jacobian of a map $v \mapsto u$ expressed in those co-ordinates, under which *only respective dimensions interact* ($\frac{\partial u_i}{\partial v_j} = 0$, $i \neq j$).

**Singular Vector Paths and Seams**: The directional derivative $\boldsymbol{J}_z \boldsymbol{v}^i = \boldsymbol{U}\boldsymbol{S}\boldsymbol{V}^\top \boldsymbol{v}^i = s^i \boldsymbol{u}^i$ shows that a small perturbation by right singular vector $\boldsymbol{v}^i$ at $z \in \mathcal{Z}$ translates under $g$ to a small perturbation in

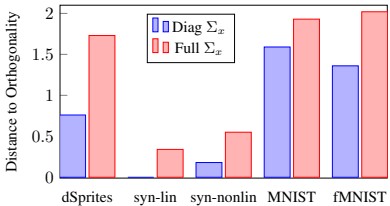 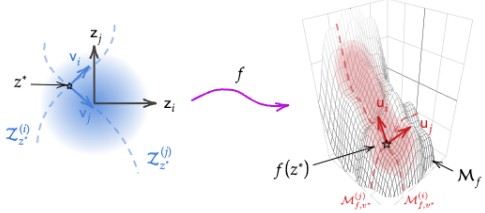

Figure 3: (*left*) **Empirical support for P1**: Rolinek et al. (2019) show that VAEs with diagonal $\Sigma_x$ have increased orthogonality in the decoder Jacobian (C1). (*right*) **Seam factorisation**: For continuous, injective, differentiable *a.e.* (c.i.d.a.e.) $g : \mathcal{Z} \to \mathcal{X}$, with Jacobian $\boldsymbol{J}_{z^*}$ and manifold $\mathcal{M}_g \subseteq \mathcal{X}$, *singular-vector paths* $\mathcal{V}_{z^*}^k \subseteq \mathcal{Z}$ (D3, dashed blue) following right singular vectors $\boldsymbol{v}^k$ of $\boldsymbol{J}_{z^*}$ (solid blue), map to *seams* $\mathcal{M}_{g,z^*}^k \subseteq \mathcal{M}_g$ (D4, dashed red) following left singular vectors $\boldsymbol{u}^k$ at $g(z^*) \in \mathcal{X}$ (solid red). $\boldsymbol{z}_k \in \mathcal{Z}$ are standard basis vectors.

direction $\boldsymbol{u}^i$ at $g(z) \in \mathcal{M}_g$. By extension, if a path in $\mathcal{Z}$ follows $\boldsymbol{v}^i$ at each point (as a vector field) its image on $\mathcal{M}_g$ is expected to be a path following $\boldsymbol{u}^i$. Note that wherever the Jacobian is continuous, each of its SVD components is continuous. Since columns of an SVD can be validly permuted, their order must be fixed for paths over "$i$-th" singular vectors to be well defined:

**Definition D2** (**Regular set and continuous SVD**). *For* c.i.d.a.e. $g : \mathcal{Z} \to \mathcal{X}$, *define the* regular set

$$\mathcal{Z}_{\text{reg}} \doteq \left\{ z \in \mathcal{Z} \;\middle|\; \boldsymbol{J}_z \text{ exists, has full column rank, and } s^1(z) > \cdots > s^d(z) > 0 \right\}.$$

*Vector fields $z \mapsto \boldsymbol{v}^i(z)$, $z \mapsto \boldsymbol{u}^i(z)$ and singular values $z \mapsto s^i(z)$ can be made continuous on each connected component of $\mathcal{Z}_{\text{reg}}$ by fixing the SVD $\boldsymbol{J}_z = \boldsymbol{U}_z \boldsymbol{S}_z \boldsymbol{V}_z^\top$.*[3][4]

With this, we define paths following $i$-th singular vectors: **singular vector paths**, or **s.v. paths**, $\mathcal{V}_z^i$ follow $\boldsymbol{v}^i$ in $\mathcal{Z}$ (Def. 3, blue dashed lines in Fig. 3); and **seams**, $\mathcal{M}_{g,z}^i$ follow $\boldsymbol{u}^i$ over the manifold (Def. 4, red dashed lines in Fig. 3) By construction, $g$ maps s.v. paths to seams (proved in Lemma C.1).

Singular vector paths and seams naturally extend the notion that right singular vector perturbations map to *distinct* left singular vector perturbations (since $\boldsymbol{S}$ is diagonal). Key to disentanglement is how 1-D densities over s.v. paths push-forward under $g$ to 1-D densities on the manifold $\mathcal{M}_g$:

**Lemma 5.1** (**Factorisation over seams**). *Let $g : \mathcal{Z} \to \mathcal{X}$ be* c.i.d.a.e. *and the prior factorise as $p(z) = \prod_{i=1}^d p_i(z_i)$ (e.g. standard Gaussian). Then, the manifold density $p_\mu$ on $\mathcal{M}_g$ factorises as*

$$p_\mu\big(g(z)\big) \;=\; \prod_{i=1}^d \frac{p_i(z_i)}{s^i(z)}\,, \qquad\qquad \textit{for every } z \in \mathcal{Z}_{\text{reg}}. \qquad (7)$$

*Moreover, each factor $\frac{p_i(z_i)}{s^i(z)}$ is the 1-D density over the $i$-th seam $\mathcal{M}_{g,z}^i$ at $x = g(z)$, obtained by pushing forward the 1-D marginal $p_i(z_i)$ over $\mathcal{V}_z^i$, the $i$-th s.v. path though $z$.*

*Proof.* See Appendix C. Follows from the standard change–of–variables formula.

**Summary**: Singular paths in $\mathcal{Z}$ are *the* latent curves along which $g$ changes in the corresponding seam on $\mathcal{M}_g$ (Lemma C.1). Lemma 5.1 states that $p_\mu$ decomposes as a product of 1-D densities along seams, each the push–forward of the marginal over the $i$-th latent s.v. path – precisely the *factorisation* condition required for disentanglement (Eq. 5). Disentanglement also requires that s.v. paths are *axis-aligned* (in general they may curve, as in Fig. 3) and that factors are *independent*, i.e. only factor $i$ changes over seam $i$. We now show that these extra properties follow from **C1** and **C2**.

**Theorem 5.2** (**Disentanglement $\Leftrightarrow$ C1-C2**). *Let $g : \mathcal{Z} \to \mathcal{X}$ be* c.i.d.a.e. *and the prior factorise as $p(z) = \prod_{i=1}^d p_i(z_i)$ (e.g. standard Gaussian). The push-forward density $p_\mu$ on the manifold $\mathcal{M}_g = \{g(z)\}$ is disentangled (D1) if and only if $g$ satisfies **C1** and **C2** almost everywhere.*

*Proof.* See Appendix D.1

Thm. 5.2 means that **C1-C2** are *precisely* the constraints needed for the factorisation in Eq. 7 to become disentanglement: **C1** causes s.v. paths to axis-align and $p_i(z_i)$ to be independent; **C2** rules out $s^i$ (of factor $i$) varying in co-ordinates $z_{j \neq i}$, ensuring seam factors are independent (see Fig. 4).

---

[3]e.g. start from an arbitrary point, order singular values strictly decreasing and choose signs continuously.

[4]Restricting to $\mathcal{Z}_{\text{reg}}$ only excludes points where $\boldsymbol{J}_z$ is undefined or has repeated singular values; these edge cases can be avoided without affecting the results. All statements are made on a fixed connected component of $\mathcal{Z}_{\text{reg}}$.

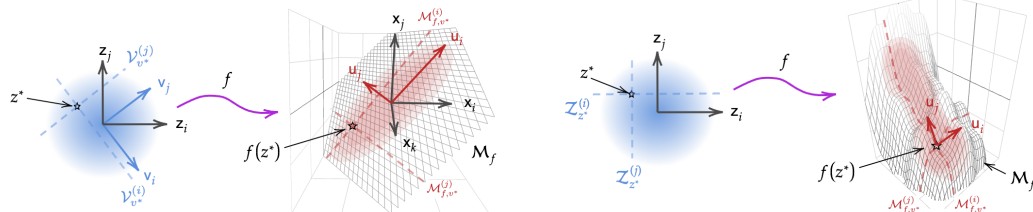

Figure 4: **Pushing forward $p(z)$, from singular vector paths to seams**: 1-D marginals $p_i(z_i)$ over *s.v. paths* $\mathcal{V}^i_{z^*}$ (dashed blue) factorise $p(z)$; and push-forward to 1-D seam densities over *seams* $\mathcal{M}^i_{f,z^*}$ (dashed red) that factorise $p_\mu$ (Lemma 5.1). (***left***) For linear $f$ *without* **C1** (e.g. full-$\Sigma_x$ LVAE), $\mathcal{V}^i_{z^*}$ are straight lines but need not axis-align (as shown in Fig. 2, *right*). (***right***) For c.i.d.a.e. $f$ satisfying **C1-C2** (e.g. Gaussian VAE under P1), $\mathcal{V}^i_{z^*}$ are axis-aligned (by C1) and seam densities are independent components (by C2) that factorise $p_\mu$. (Criteria for disentanglement (D1) are underlined.)

Thus, to the extent a Gaussian VAE with diagonal posteriors induces property P1 by Eq. 6, it expressly disentangles the decoder's push-forward density; and to the extent disentanglement is observed, diagonalisation constraints **C1-C2** must hold. This provides a firm justification for how disentanglement emerges in VAEs, while the relationship between Eq. 6 and constraints C1-C2 also suggests a plausible rationale for why disentanglement arises inconsistently (Locatello et al., 2019).

## 6 IDENTIFIABILITY

We now investigate if a model, capable of fitting the data (i.e. data is generated under the model class), learns the *true* generative factors up to some symmetry, or could settle on a spurious factorisation.

**Corollary 6.1** (**LVAE Identifiability**). *Let data be generated under the linear Gaussian LVM Eq. 2 with ground-truth $g(z) = Wz$, $W = U_w S_w V_w^\top \in \mathbb{R}^{m \times d}$ of full column rank and* distinct *singular values. Let an LVAE with* diagonal *posteriors be trained on $n$ samples, and as $n \to \infty$ its learned parameters yield $p_\mu^{(d)} \equiv p_\mu^{(g)}$ on the mean manifold. Then the LVAE achieves disentanglement (D1) and identifies ground-truth independent components on $\mathcal{M}_g$ up to permutation and sign (**P&S**).*

*Proof.* See Appendix D.2 (Follows from the uniqueness of the SVD). $\square$

Thus, if an LVAE learns to model the data, it learns the ground truth independent factors.

*Remark* 6.2 ($V_w$ **immaterial**). Ground-truth right singular vectors $V_w$ are not recoverable from $p(x)$ under the PPCA/LVAE model; this *is not* a lack of identification. With a standard Gaussian prior, any orthonormal change of basis of $z$ preserves independence and leaves $p(x)$ unchanged. The only data-relevant object is $U_w S_w$; the arbitrary basis in which $W$ was written has no bearing on $p(x)$.

The linear case hints at why independent factors may be identifiable more generally, since it depends on the Jacobian SVD, fundamental to the non-linear case. Taking this hint, we show that if a manifold density admits a seam factorisation (as in Lemma 5.1), that seam factorisation is unique (P&S) and *intrinsic* to $p_\mu$, agnostic to any generative process or parameterisation. It follows that if the push-forward of a decoder fits $p_\mu$, then its seams must align with the intrinsic seams of $p_\mu$ (P&S); and subsequently that a Gaussian VAE fitting $p_\mu$ *identifies ground truth factors* (P&S). (Proofs in D.3)

**Lemma 6.3** (**Seams are Intrinsic**). *Let $\mathcal{M} \subseteq \mathcal{X}$ carry a manifold density $p_\mu$. Assume that on a regular set $\mathcal{M}_{\text{reg}}$ there exist scalar functions $\{u_i(x)\}_{i=1}^d$ (each varying only along a 1-D curve through $x$, i.e. a* seam*) and 1-D densities $\{f_i\}$ such that*

$$p_\mu(x) = \prod_{i=1}^d f_i\big(u_i(x)\big), \qquad x \in \mathcal{M}_{\text{reg}}, \tag{8}$$

*and that the on-manifold Hessian $\mathbf{H}_x \doteq \nabla_x^2 \log p_\mu(x)$ has pairwise distinct eigenvalues a.e. on $\mathcal{M}_{\text{reg}}$. Then for each $x \in \mathcal{M}_{\text{reg}}$, the $d$ seam directions (along which exactly one $u_i$ varies) are determined intrinsically by $p_\mu$, as eigenvectors of $\mathbf{H}_x$, unique up to permutation and sign (P&S).*

**Lemma 6.4 (A matching decoder finds seams).** *Under assumptions of Lemma 6.3, let $d : \mathcal{Z} \to \mathcal{X}$ be* c.i.d.a.e. *with factorised prior $p(z) = \prod_i p_i(z_i)$ and push-forward density $p_\mu^{(d)} \equiv p_\mu$ matching on $\mathcal{M}_d \doteq \{d(z)\} = \mathcal{M}$. If $d$ satisfies C1–C2 a.e., then for any $z$ and $x = d(z)$:*

- *left singular vectors $U_z$ of $J_z$ coincide with the seam directions in Lemma 6.3 (up to P&S);*
- *the images under $d$ of singular-vector paths (D3) are exactly the seams through $x$;*
- *along the $i$-th seam, the factor $f_i$ is the 1-D push-forward of $p_i(z_i)$ (as in Lemma 5.1).*

**Theorem 6.5 (Gaussian VAE Identifiability).** *Let data be generated by* c.i.d.a.e. *$g : \mathcal{Z} \to \mathcal{X}$ with factorised prior $p(z) = \prod_{i=1}^d p_i(z_i)$. Let a Gaussian VAE with* diagonal *posteriors learn a decoder $d : \mathcal{Z} \to \mathcal{X}$. Suppose both $g$ and $d$ satisfy C1–C2 and manifold densities match: $p_\mu^{(d)} \equiv p_\mu^{(g)}$ on $\mathcal{M} = \{g(z)\} = \{d(z)\}$. If, eigenvalues of the tangent Hessian (see proof) are pairwise distinct a.e., then $d$ identifies ground-truth independent components on $\mathcal{M}_g$, up to permutation and sign (P&S).*

Thus, if a Gaussian VAE fits the push-forward of a Gaussian distribution under the conditions of 5.2, then the VAE identifies and disentangles the ground truth generative factors (up to permutation/sign).

*Remark* 6.6. P&S symmetry is optimal since seams follow $u_i$, with no inherent order or orientation.

We can also consider fitting a Gaussian VAE to data sampled from the push-forward of other priors.

**Corollary 6.7 (BSS).** *In Theorem 6.5, if priors $p^{(g)}(z)$ and $p^{(d)}(z)$ factorise and $p_\mu^{(g)} \equiv p_\mu^{(d)}$ with C1-C2 holding a.e., then the seam decomposition $p_\mu$ on $\mathcal{M}$ is unique up to permutation and sign, and $g$ and $p^{(g)}(z)$ are recoverable up to an axis-aligned diffeomorphism $\phi \doteq g^{-1} \circ d : \mathcal{Z} \to \mathcal{Z}$.*

*Proof.* Immediate from proof of Theorem 6.5, which does not depend on the form of $p(z)$. The diffeomorphism follows since $\mathcal{M}_g = \mathcal{M}_d$ and by injectivity of $d$, $g$. □

*Remark* 6.8 (**Gaussian $p(z)$ unidentifiability**). Classical non-linear ICA aims to identify ground truth factors of the model in Theorem 6.5 with deterministic $p_\theta(x|z) = \delta_{x-d(z)}$, which is impossible if $p(z)$ is Gaussian (Khemakhem et al., 2020; Locatello et al., 2019). Corollary 6.1 and Theorem 6.5 show that under a probabilistic formulation together with constraints C1–C2 induced by diagonal posteriors, we obtain identifiability without requiring extra side information, e.g. auxiliary variables.[5]

In summary, we have defined disentanglement, shown that it holds if and only if C1-C2 hold, and that seam factors are unique and therefore identifiable, up to expected symmetry.

## 7 EMPIRICAL SUPPORT

We include empirical results to illustrate disentanglement and support our claims. From our analysis, we expect (i) diagonal posteriors to promote diagonalised derivative terms; and (ii) diagonalised derivatives to correlate with disentanglement.

Both (i) and (ii) are well illustrated in the linear case where ground truth factors are known analytically. Fig. 2 shows results for diagonal and full covariance LVAEs learning Gaussian parameters (see caption for details). All models learn optimal parameters as expected (*left*); but, as (i) predicts, only diagonal covariances cause right singular vectors of $J_z$ to converge to standard basis vectors, $V \to I$ (C1), (*centre*); hence latent traversals map to independent components along left singular vectors $u^i$ (*right*), yielding disentanglement (D1), predicted by (ii). This evidences diagonal covariances "breaking the rotational symmetry" of a Gaussian prior. While the linear case seems trivial, it is a fundamental demonstration since our analysis shows that disentanglement in general follows the same rationale. Interestingly Fig. 3 shows that learning parameters a disentangled model is notably slower (*left*), due to the rate at which $V \to I$ (*centre*). A diagonal covariance model must find one of a finite set of solutions among the infinite solutions of a full covariance model.

Various studies show further empirical support. Supporting (i) and (ii), Rolinek et al. (2019) show that columns of a decoder's Jacobian are more orthogonal (i.e. $V \to I$, C1) in VAEs with diagonal posterior than those with full posteriors (Fig. 3, *left*), and that diagonality correlates with disentanglement. Supporting (ii), Kumar & Poole (2020) show that directly inducing column-orthogonality in the decoder Jacobian promotes disentanglement.

---

[5]Unidentifiability proofs typically make use of an arbitrary rotation applied to the Gaussian prior (*cf* $R$ in Eq. 3), but we have seen that C1 removes such symmetry, even in the linear case (Fig. 2). (See also Remark 6.2.)

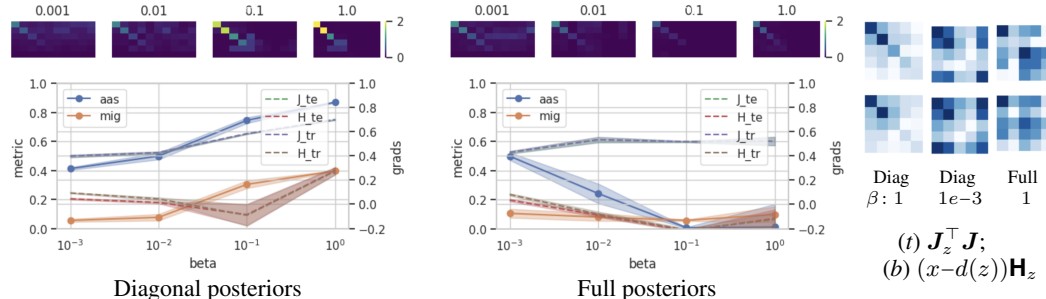

Figure 5: **Diagonal vs Full Posteriors**: (*left*) (*bottom*) disentanglement metrics and estimated diagonality of Eq. 6 terms (see Appendix E). With diagonal posteriors, disentanglement and diagonality are correlated (supporting P1), relative to full posteriors; (*top*) heatmaps of mutual information between model latents and ground truth factors. (*right*) derivatives in Eq. 6. terms are less diagonal for lower beta or full posteriors. (All results averaged over multiple runs)

Complementing prior work, we train diagonal and full posterior Gaussian VAEs ($d = 10$) on the *dSprites* dataset, for which the 5 ground truth generative factors are known (all results averaged over 5 runs). How well each latent co-ordinate identifies a ground truth factor can be estimated explicitly from their *mutual information*, and a function of mutual information often captures a model's overall disentanglement, e.g. the *mutual information gap* (MIG, Chen et al. (2018)). Fig. 5 (*main plots*) reports MIG, *axis alignment score* (AAS) (a novel metric based on the entropy of the mutual information distribution) and derivative diagonality estimates against $\beta$ (see Appendix E for details). For diagonal posterior VAEs (*left*), disentanglement and diagonality both broadly increase with $\beta$. For full posteriors (*right*) no clear trend is observed. The heatmaps above (aligned with plots by $\beta$) show mutual information between each latent co-ordinate and ground truth factor (ordered greedily to put highest mutual information scores along the diagonal). We see that for diagonal posterior VAEs, disentanglement increases with $\beta$ and that individual latent co-ordinates (horizontal) correlate with distinct ground truth factors (vertical), whereas that trend is not observed for full posteriors. For illustration, Fig. 5 (right) shows heatmaps of the $d \times d$ derivative terms in Eq. 6 ($\boldsymbol{J}_z^\top \boldsymbol{J}_z$ term top, Hessian term bottom), each for diagonal covariances, $\beta = 1$ (*l*); diagonal covariances, $\beta = 0.001$ (*c*); and full covariances, $\beta = 1$ (*r*) (each averaged over a batch). We perform comparable analysis on the CelebA dataset of natural face images, reported in Appendix F.

Our understanding of the interplay between $\beta$ and disentanglement is that higher $\beta$ implies higher expected noise, weakening reconstructions but enhancing disentanglement; while lower $\beta$ (less assumed noise) tightens reconstructions but limits disentanglement to concentrated, potentially disconnected regions of $\mathcal{Z}$ (§4). This suggests that the heuristic of starting with $\beta$ high and reducing it over training may give both disentangled and higher quality samples. We run experiments on the dSprites dataset and report results in Appendix G for constant $\beta$ baselines (1, 0,001), and exponentially reducing $\beta$ ($1 \rightarrow 0.001$) over training. The results are as expected results, showing that annealing $\beta$ gives both sharp reconstructions and good disentanglement. Noting also the resemblance to the "de-noising" process in denoising autoencoders and diffusion models, this suggests that dynamically varying $\beta$ is an interesting direction for future research.

# 8 RELATED WORK

Higgins et al. (2017) showed that disentanglement in VAEs is enhanced by setting $\beta > 1$ in the ($\beta$-)ELBO (Eq. 1). Burgess et al. (2018) conjectured that diagonal posterior covariances may cause disentanglement. Rolinek et al. (2019) showed supporting empirical evidence (Fig. 3) and derived an approximate relationship between diagonal posteriors and Jacobian orthogonality, conjectured to then cause disentanglement. Kumar & Poole (2020) generalised the argument, reaching an approximation to the identity in Eq. 6. We make the link between posterior covariances and decoder derivatives precise in Eq. 6 and, by giving disentanglement a formal definition, show how it follows from Eq. 6 via constraints C1-C2, confirming the conjecture.

Lucas et al. (2019); Bao et al. (2020) and Koehler et al. (2022) study properties of linear VAEs. Notably Lucas et al. (2019) show the equivalence of $\beta$ and Var$[x|z]$ in Gaussian VAEs, which we generalise in Appendix B; and prove identifiability of LVAEs, which we generalise to the non-linear case. Zietlow et al. (2021) show that disentanglement can be sensitive to perturbing the data. Reizinger

et al. (2022) seek to relate the ELBO to *independent mechanism analysis* (Gresele et al., 2021), which encourages column-orthogonality in the mixing function of ICA.[6] We show that Jacobian orthogonality (**C1**) is insufficient for disentanglement/identification of independent components, which also requires (**C2**). VAEs relate closely to ICA (§2) and noisy ICA (Hyvarinen, 1998). The latter assumes the same generative model but does not model the posterior, which we show is the critical factor for disentanglement.

Ramesh et al. (2018) trace independent factors by following leading left singular vectors (our *seams*) of the Jacobian of a GAN generator, whereas Chadebec & Allassonnière (2022) and Arvanitidis et al. (2018) consider paths in latent space defined by the inverse image of paths over the data manifold (our *s.v. paths*). Pan et al. (2023) claim that the data manifold is identifiable from a geometric perspective assuming Jacobian-orthogonality, differing to our probabilistic factorisation approach. Bhowal et al. (2024) consider linear and non-linear components of the encoder/decoder, loosely resembling our Jacobian SVD view. However, dissecting a function into linear/non-linear components is not well defined, whereas the SVD is unique (up to permutation/sign). Buchholz et al. (2022); Buchholz & Schölkopf (2025) analyse identifiability by function classes, e.g. proving that *conformal maps* are identifiable and *orthogonal coordinate transformations* (satisfying C1) are not. By comparison, our C1-C2 are derived with respect to disentanglement, which implies an intrinsic density factorisation, unique/identifiable up to symmetry (Thm 6.5). Brady et al. (2023); Lachapelle et al. (2023) analyze *additive/compositional* decoders that have block-diagonal Hessians in pixel space, a strictly stronger condition than our Thm 5.2, which allows, e.g., rotations/scale/colour changes that would not be block-diagonal in pixel space.

## 9 CONCLUSION

Unsupervised disentanglement of generative factors of the data is of fundamental interest in machine learning. Thus, irrespective of current popularity, understanding how a VAE disentangles the data *for free* may offer useful insight for other paradigms. We take significant strides in this respect, in particular proposing a simple, formal definition of disentanglement (D1) as factorising the manifold density into independent components, each factor being the image of an axis-aligned traversal in latent space. We also give a simple interpretation of $\beta$ in a $\beta$-VAE, as adjusting the assumed variance of the likelihood (generalising the known Gaussian case), justifying both why $\beta > 1$ promotes disentanglement while degrading generative quality, and $\beta < 1$ mitigates *posterior collapse*.

Our key results (Definition 1, Lemma 5.1, Theorem 5.2, Lemma 6.3, Lemma 6.4, Theorem 6.5) are **not specific to VAEs**, but are general to smooth push-forwards of a factorised prior, as also in GANs and flows. We show via a relatively simple mechanism, the decoder Jacobian's SVD, that under suitable conditions a push-forward density factorises. Indeed the factorisation in the latent space can be seen to project *en masse* onto the manifold, whereby independent densities over *singular vector paths* in latent space push-forward to independent densities over *seams* on the manifold. We show that the constraints needed for such a factorisation to reach disentanglement are precisely those imposed, in aggregate and in expectation, by a VAE with diagonal posteriors, justifying both why disentanglement arises and why it is also ephemeral (Locatello et al., 2019). Furthermore, independent factors are provably *identifiable*, which is particularly significant given their proven *unidentifability* under ICA with Gaussian prior.

Neural networks models are often considered too complex to explain, despite their increasingly widespread deployment in everyday applications. An improved theoretical understanding seems essential to optimally and safely take advantage of machine learning progress, particularly in critical systems. We hope our work is a useful step in that direction, providing new insight into how the data density decomposes over independent generative factors. Interestingly, our proof structure shows that, irrespective of the complex non-linearity of a decoder, how the prior pushes forward can be considered relatively simply.

VAEs (and variants e.g. AEs, SAEs) form part of a pipeline in many state-of-the-art models, e.g. latent diffusion (e.g. Rombach et al., 2022; Pandey et al., 2022; Yang et al., 2023; Zhang et al., 2022) and LLMs; other recent works show that supervised learning (Dhuliawala et al., 2024) and self-supervised learning (Bizeul et al., 2024) can be viewed as latent models trained under ELBO variants. In future work we will look to see if our results transfer to such other learning paradigms.

---

[6]We report apparent discrepancies in Reizinger et al. (2022) in Appendix H.

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

## A    PROOF OF DECODER DERIVATIVE CONSTRAINTS

**Property P1.** *$\boldsymbol{J}_z^\top \boldsymbol{J}_z$ and $(x-d(z))^\top \mathsf{H}_z$ in Eq. 6 are each diagonal for $z$ concentrated around $\mathbb{E}[z|x]$.*

**Lemma 4.1** (**Disentanglement constraints**). *For a trained Gaussian VAE and $x, z$ satisfying P1:*

**C1)** *Right singular vectors $\boldsymbol{V}_z$ of the decoder Jacobian $\boldsymbol{J}_z$ are standard basis vectors, i.e. after relabeling/sign flips of the latent axes, we have $\boldsymbol{V}_z = \boldsymbol{I}$;*

**C2)** *The matrix of partial derivatives of singular values $(\frac{\partial s_i}{\partial z_j})_{i,j}$ is diagonal, i.e. $\frac{\partial s_i}{\partial z_j}=0$ for all $i \neq j$.*

*Proof. (**Preliminaries**)*:   Recall $p(z) = \mathcal{N}(0, I)$ and $p(x \mid z) = \mathcal{N}(x; d(z), \sigma^2 \boldsymbol{I})$. Let $q(z \mid x) = \mathcal{N}(z; e(x), \Sigma_x)$ be the trained posterior with $\Sigma_x$ diagonal (by assumption). Denote the SVD of the decoder Jacobian by

$$\boldsymbol{J}_z \; = \; \boldsymbol{U}_z \, \boldsymbol{S}_z \, \boldsymbol{V}_z^\top, \qquad \boldsymbol{U}_z \in \mathbb{R}^{n \times d}, \quad \boldsymbol{S}_z = \mathrm{Diag}(s_1(z), \ldots, s_k(z)), \quad s_i(z) > 0.$$

Assume full column rank on the manifold ($s_i(z) > 0$). For Gaussian likelihood with variance $\sigma^2$, the Hessian of the log-likelihood w.r.t. $z$ can be written

$$\nabla_z^2 \log p(x \mid z) \; = \; -\tfrac{1}{\sigma^2} \big( \boldsymbol{J}_z^\top \boldsymbol{J}_z - \sum_{\ell=1}^n r(z)_\ell \, \boldsymbol{H}_\ell(z) \big),$$

where $r(z) = x - d(z)$ and $\boldsymbol{H}_\ell(z) \in \mathbb{R}^{k \times k}$ is the Hessian of the $\ell$-th decoder coordinate, $[\boldsymbol{H}_\ell]_{pq} = \partial^2 d_\ell / \partial z_p \, \partial z_q$. Combined with the Opper–Archambeau fixed-point yields[7]

$$\Sigma_x^{-1} \; = \; \boldsymbol{I} - \mathbb{E}_q\big[\nabla_z^2 \log p(x \mid z)\big] \; = \; \boldsymbol{I} + \tfrac{1}{\sigma^2}\mathbb{E}_q[\boldsymbol{J}_z^\top \boldsymbol{J}_z] \; - \; \tfrac{1}{\sigma^2}\mathbb{E}_q\big[\sum_{\ell=1}^n r(z)_\ell \, \boldsymbol{H}_\ell(z)\big]. \quad (9)$$

(**C1**):   For diagonal $\Sigma_x$ in Eq. 9 and $z$ concentrated around $e(z)$ under P1 ("no cancellation"), we have

$$\boldsymbol{J}_z^\top \boldsymbol{J}_z \; \text{is diagonal}, \qquad \sum_{\ell=1}^n r(z)_\ell \, \boldsymbol{H}_\ell(z) \; \text{is diagonal}. \quad (10)$$

Diagonal $\boldsymbol{J}_{e(x)}^\top \boldsymbol{J}_{e(x)} = \mathrm{Diag}(s_1^2, \ldots, s_k^2)$ implies right singular vectors $\boldsymbol{v}^i$ are the standard basis, up to signed permutations. By relabelling latent axes and absorbing signs, $\boldsymbol{V}_z = \boldsymbol{I}$ for all $z$ visited by the encoder, establishing **C1**.

(**Directed Hessian is Tangent to Manifold**):   Since the model is well trained, we assume $d(e(x)) \approx x$. For $z = e(x) + \delta$ with $\delta > 0$ small, a first-order Taylor expansion gives

$$d(z) \; = \; d(e(x)) + \boldsymbol{J}_{e(x)} \, \delta + O(\|\delta\|^2) \qquad \Longrightarrow \qquad r(z) = x - d(z) \; = \; -\boldsymbol{J}_{e(x)} \, \delta + O(\|\delta\|^2).$$

Thus, for $z$ concentrated around $e(x)$, to first order, $r(z)$ lie in the column space of $\boldsymbol{J}_{e(x)}$, i.e. in the span of the left singular vectors $\{u_i(e(x))\}_{i=1}^k$ (columns of $\boldsymbol{U}_z$):

$$r \; = \; \boldsymbol{U}_z a, \qquad a \in \mathbb{R}^k \quad (11)$$

Hence, we consider $\sum_{\ell=1}^n r_\ell \, \boldsymbol{H}_\ell(z)$ to be diagonal for all $r \in \mathrm{span}(\boldsymbol{U}_z)$. In particular, $\sum_{\ell=1}^n u_{i\ell}^\top \boldsymbol{H}_\ell(z)$ is diagonal for all rows $u_i$ of $\boldsymbol{U}_z$ and, by definition of slices $\boldsymbol{H}_\ell$ and $\boldsymbol{J}_z$,

$$\big[\sum_{\ell=1}^n u_{i\ell}^\top \boldsymbol{H}_\ell(z)\big]_{pq} \; = \; \sum_{\ell=1}^n u_{i\ell}^\top \frac{\partial^2 d_\ell}{\partial z_p z_q} \; = \; \big(\boldsymbol{U}_z^\top \frac{\partial \boldsymbol{J}_z}{\partial z_p}\big)_{iq} \quad (12)$$

(**Diagonality of** $(\frac{\partial s_i}{\partial z_j})_{i,j}$):

Differentiating $\boldsymbol{U}_z^\top \boldsymbol{U}_z = \boldsymbol{I}_k$ to get $\frac{\partial \boldsymbol{U}_z^\top}{\partial z_j}\boldsymbol{U}_z + \boldsymbol{U}_z^\top \frac{\partial \boldsymbol{U}_z}{\partial z_j} = \boldsymbol{0}$ shows $\Omega_j(z) := \boldsymbol{U}_z^\top \frac{\partial \boldsymbol{U}_z}{\partial z_j} \in \mathbb{R}^{d \times d}$ is skew-symmetric. Differentiating $\boldsymbol{J}_z = \boldsymbol{U}_z \, \boldsymbol{S}_z$ w.r.t. $z_j$ and premultiplying by $\boldsymbol{U}_z^\top$ then gives

$$\boldsymbol{U}_z^\top \frac{\partial \boldsymbol{J}_z}{\partial z_j} \; = \; \Omega_j(z) \, \boldsymbol{S}_z \; + \; \frac{\partial \boldsymbol{S}_z}{\partial z_j}. \quad (13)$$

---

[7]we use: $\Sigma_x^{-1} = -\mathbb{E}_q[\nabla_z^2 \log p(x, z)] = -\mathbb{E}_q[\nabla_z^2 \log p(z) + \nabla_z^2 \log p(x|z)]; \quad \nabla_z^2 \log p(z) = -\boldsymbol{I}.$

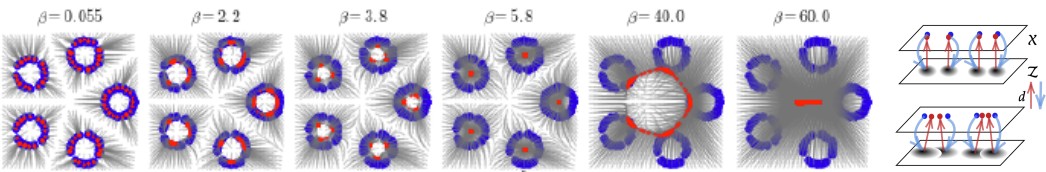

Figure 6: Illustrating $\beta \propto \mathrm{Var}[x|z]$ (blue=data, red=reconstruction): (*l*) For low $\beta$ ($\beta=0.55$), $\mathrm{Var}[x|z]$ is low (by Eq. 6), and data must be well reconstructed (right, top). As $\beta$ increases, $\mathrm{Var}[x|z]$ and so $\mathrm{Var}[z|x]$ increase, and posteriors of nearby samples $\{x_i\}_i$ increasingly overlap (right, bottom). For $z$ in overlapping $\{q(z|x_i)\}_i$, the decoder $\mathbb{E}[x|z]$ maps to a weighted average of $\{x_i\}_i$. Initially, close neighbours reconstruct to their mean ($\beta=2.2, 3.8$), then small circles "become neighbours" and map to their centres. Finally ($\beta=60$), all samples reconstruct to the global centroid. (reproduced with permission from Rezende & Viola, 2018) (*r*) illustrating posterior overlap, (*t*) low $\beta$, (*b*) higher $\beta$.

Since $\Omega_j(z)$ is skew-symmetric, all diagonal entries are zero; since $\frac{\partial \boldsymbol{S}_z}{\partial z_j}$ is diagonal, all non-diagonal entries are zero. Thus, of respective entries, only one is non-zero and can be considered separately.

From Eq. 12, only $(\Omega_j(z)\,\boldsymbol{S}_z)_{:j}$ elements can be non-zero, thus *all* elements of $\Omega_j(z)$ must be zero (by skew-sym.), **ruling out rotation in the tangent plane**[8]

For $\frac{\partial \boldsymbol{S}_z}{\partial z_j}$, diagonality (only $(\frac{\partial \boldsymbol{S}_z}{\partial z_j})_{kk}$ elements non-zero) and Eq. 12 (only $(\frac{\partial \boldsymbol{S}_z}{\partial z_j})_{:j}$ elements non-zero) imply that only elements $(\frac{\partial \boldsymbol{S}_z}{\partial z_j})_{jj} = \frac{\partial s_j}{\partial z_j}$ can be non-zero, eliminating mixed partials

$$\frac{\partial s_i}{\partial z_j}(z) \;=\; 0 \quad \text{for all } i \neq j,$$

i.e. the Jacobian of the singular-value map $s(z) = (s_1(z), \ldots, s_k(z))$ is *diagonal*, proving **C2**. $\qquad\square$

# B   $\beta$ CONTROLS NOISE VARIANCE

Choosing $\beta > 1$ in Eq. 1 can enhance disentanglement (Higgins et al., 2017; Burgess et al., 2018) and has been viewed as re-weighting ELBO components or as a Lagrange multiplier. We show that $\beta$ implicitly controls the likelihood's variance and that the "$\beta$-ELBO" remains a valid objective.

Dividing the ELBO by a constant and suitably adjusting the learning rate leaves the VAE training algorithm unchanged, hence consider Eq. 1 divided through by $\beta$ with the log likelihood scaled by $\beta^{-1}$. For a Gaussian VAE with $\mathrm{Var}[x|z] = \sigma^2$, this exactly equates to a standard VAE with variance $\beta\sigma^2$ (Lucas et al., 2019). More generally, scaling the log likelihood by $\beta^{-1}$ is equivalent to an *implicit likelihood* $p_\theta(x|z)^{1/\beta}$, where $\beta$ acts as a *temperature* parameter: $\beta \to \infty$ increases the effective entropy towards uniform (the model assumes more noise in the data, fitting more loosely), and $\beta \to 0$ reduces it to a delta (reconstructions should be tight). Optimal posteriors fit to the implicit likelihood, $q_\phi(z|x) \propto p_\theta(x|z)^{1/\beta} p(z)$, which thus dilate ($\beta > 1$) or concentrate ($\beta < 1$). This generalises the Gaussian result (Lucas et al., 2019), showing that the $\beta$-ELBO is simply the ELBO for a different likelihood model.[9]

**Empirical support**: Our claim, in effect that $\mathrm{Var}[x|z] \propto \beta$, is well illustrated on synthetic data in Fig. 6 (Rezende & Viola, 2018, see caption for details). It also immediately explains *blur in $\beta$-VAEs* since $\beta > 1$ simply assumes more noise. It also explains why $\beta < 1$ helps mitigate *posterior collapse* (Bowman et al., 2015), i.e. when a VAE's likelihood is sufficiently expressive that it can directly model the data distribution, $p(x|z) = p(x)$, leaving latent variables redundant (posterior "collapses" to prior). As $\beta \to 0$, the effective variance of $p_\theta(x|z)$, and the distributions it can describe, reduces. Thus for some $\beta < 1$ the effective variance falls below $\mathrm{Var}[x]$, rendering posterior collapse impossible as some variance in $x$ can only be explained by $z$. Thus our claim that $\beta$ controls effective variance explains well-known empirical observations, which in turn provide empirical support for the claim.

---

[8]we have: $\Omega_j(z)_{kj} = -\Omega_j(z)_{jk} = 0$, if $j \neq k$; and $\Omega_j(z)_{jj} = 0$ (skew-sym.).

[9]Technically, the $\beta$-ELBO's value is incorrect without renormalising the implicit likelihood, but that is typically irrelevant, e.g. for commonly used Gaussian likelihoods, only the quadratic "MSE" term appears in the loss.

## C  Appendix: Singular Vector Paths and Seams

**Definition D3 ($i$-th singular–vector path).** *Let $g : \mathcal{Z} \to \mathcal{X}$ be c.i.d.a.e.. For $z^* \in \mathcal{Z}_{\mathrm{reg}}$, $i \in \{1, \dots, d\}$, the $i$-th singular–vector path (s.v. path) through $z^*$ is any $C^1$ curve $t \mapsto z_t^i$ with $z_0^i = z^*$ satisfying*

$$\tfrac{d}{dt} z_t^i \;=\; \boldsymbol{v}^i(z_t^i) \qquad \textit{for } t \textit{ in its maximal interval } I_{z^*,i} \subseteq \mathbb{R}.$$

*We denote the path set by $\mathcal{V}_{z^*}^i \doteq \{z_t^i : \; t \in I_{z^*,i}\} \subseteq \mathcal{Z}_{\mathrm{reg}}.$[10] (See Fig. 3, left, dash blue lines).*

**Definition D4 ($i$-th seam).** *Let $g : \mathcal{Z} \to \mathcal{X}$ be c.i.d.a.e. with manifold $\mathcal{M}_g = \{g(z)\}$. For $z^* \in \mathcal{Z}_{\mathrm{reg}}$, $i \in \{1, \dots, d\}$, the $i$-th seam through $g(z^*)$ is any $C^1$ curve $t \mapsto x_t^i$ in $\mathcal{M}_g$ with $x_0^i = g(z^*)$ satisfying*

$$\tfrac{d}{dt} x_t^i \;=\; s^i\big(g^{-1}(x_t^i)\big)\, \boldsymbol{u}^i\big(g^{-1}(x_t^i)\big) \quad \textit{for } t \textit{ in } I_{z^*,i}.$$

*We denote the path set $\mathcal{M}_{g,z^*}^i \doteq \{x_t^i : \; t \in I_{z^*,i}\} \subseteq \mathcal{M}_g$ and define* seam coordinate

$$u_i(t) \;\doteq\; \int_0^t s^i\big(g^{-1}(x_\tau^i)\big)\, d\tau, \qquad so \quad \tfrac{d}{dt} u_i(t) = s^i\big(g^{-1}(x_t^i)\big), \;\; u_i(0) = 0.$$

*$u_i$ measures position along the seam in units of $s^i$ (strictly monotone as $s^i > 0$). (See Fig. 3, right).*

**Lemma C.1 (Paths $\mapsto$ seams).** *Let $g : \mathcal{Z} \to \mathcal{X}$ be c.i.d.a.e., $z^* \in \mathcal{Z}_{\mathrm{reg}}$, $i \in \{1, \dots, d\}$, and let $\mathcal{V}_{z^*}^i$ be the $i$-th s.v. path through $z^*$. Then the image of $\mathcal{V}_{z^*}^i$ under $g$ is the $i$-th seam through $g(z^*)$: $\mathcal{M}_{g,z^*}^i = \{g(z) : z \in \mathcal{V}_{z^*}^i\}$.*

*Proof.* For $x_t^i \doteq g(z_t^i)$, by the chain rule and SVD: $\frac{dx_t^i}{dt} = \boldsymbol{J}_{z_t^i} \frac{dz_t^i}{dt} = \boldsymbol{J}_{z_t^i} \boldsymbol{v}^i(z_t^i) = s^i(z_t^i)\, \boldsymbol{u}^i(z_t^i)$, so $x_t^i$ satisfies Def. 4. $\qquad\square$

**Lemma 5.1 (Factorisation over seams).** *Let $g : \mathcal{Z} \to \mathcal{X}$ be c.i.d.a.e. and the prior factorise as $p(z) = \prod_{i=1}^d p_i(z_i)$ (e.g. standard Gaussian). Then, the manifold density $p_\mu$ on $\mathcal{M}_g$ factorises as*

$$p_\mu\big(g(z)\big) \;=\; \prod_{i=1}^d \tfrac{p_i(z_i)}{s^i(z)}, \qquad \textit{for every } z \in \mathcal{Z}_{\mathrm{reg}}. \tag{7}$$

*Moreover, each factor $\frac{p_i(z_i)}{s^i(z)}$ is the 1-D density over the $i$-th seam $\mathcal{M}_{g,z}^i$ at $x = g(z)$, obtained by pushing forward the 1-D marginal $p_i(z_i)$ over $\mathcal{V}_z^i$, the $i$-th s.v. path though $z$.*

*Proof.* By a standard change–of–variables (on embedded manifolds) and $|\boldsymbol{J}_z^\top \boldsymbol{J}_z| = \prod_{i=1}^d s^i(z)^2$,

$$p_\mu\big(g(z)\big) \;=\; \det(\boldsymbol{J}_z^\top \boldsymbol{J}_z)^{-1/2}\, p(z) \;=\; \tfrac{\prod_{i=1}^d p_i(z_i)}{\prod_{i=1}^d s^i(z)}, \qquad \textit{yielding Eq. 7.}$$

For each $i$ and $x = g(z)$, the change–of–variables formula along $\mathcal{M}_{g,z}^i$ ($i$-th seam through $x$ with coordinate $u_i$, Def. 4) at $t = 0$ gives the local pushed 1-D *seam–density*

$$f_i^{(z)}\big(u_i(t)\big) \;\doteq\; \tfrac{p_i([z_t^i]_i)}{s^i(z_t^i)}, \qquad \tfrac{d}{dt} u_i(t) = s^i(z_t^i), \;\; u_i(0) = 0, \tag{14}$$

where $t \mapsto z_t^i$ is the $i$-th singular–vector path through $z$ (Def. 3) with local co-ordinate $z_i^i(t)$. Evaluating at $t = 0$: $f_i^{(z)}\big(u_i(0)\big) = \frac{p_i(z_i)}{s^i(z)}$, shows that seam-densities are the factors of Eq. 7. $\qquad\square$

---

[10] Different choices of sign for $\boldsymbol{v}^i$ reverse the time direction ($t \mapsto -t$) but generate the same path set.

# D   APPENDIX: DISENTANGLEMENT AND IDENTIFIABILITY PROOFS

## D.1   PROOF OF NON-LINEAR DISENTANGLEMENT

**Theorem 5.2** (**Disentanglement $\Leftrightarrow$ C1-C2**). *Let $g : \mathcal{Z} \to \mathcal{X}$ be c.i.d.a.e. and the prior factorise as $p(z) = \prod_{i=1}^{d} p_i(z_i)$ (e.g. standard Gaussian). The push-forward density $p_\mu$ on the manifold $\mathcal{M}_g = \{g(z)\}$ is* disentangled *(D1) if and only if $g$ satisfies **C1** and **C2** almost everywhere.*

*Proof.* (C1/2 $\Rightarrow$ D1) By Thm. 5.1, the manifold density factorises pointwise as $p_\mu(d(z)) = \prod_{i=1}^{d} \frac{p_i(z_i)}{s^i(z)}$. By **C1**, $\boldsymbol{V}_z = \boldsymbol{I}$ for all $z$, so the $i$-th singular–vector path through $z$ is exactly the axis–aligned line $\{z' : [z']_i \text{ varies}, [z']_{\neg i} = [z]_{\neg i}\}$; by Lemma C.1 its image is the $i$-th seam through $x = g(z)$ following $\boldsymbol{u}^i$. By **C2**, $s^i(z)$ depends only on $z_i$. Define the seam coordinate $u_i$ along the $i$-th seam as in D4; then $u_i$ is a strictly monotone function of $z_i$, hence the 1-D push–forward of $p_i$ along that seam is $f_i(u_i) = \left| \frac{du_i}{dz_i} \right|^{-1} p_i(z_i) = \frac{p_i(z_i)}{s^i(z_i)}$, Thus $p_\mu(g(z)) = \prod_i f_i(u_i(z))$ with each $f_i$ evaluated on the $i$-th seam. Finally, since $u_i$ is monotone in $z_i$ and $\{z_i\}$ are independent, the random variables $\{u_i\}$ are independent; hence factors $\{f_i(u_i)\}$ are *statistically independent* as required by D1.

(D1 $\Leftarrow$ C1/2) Assume $p_\mu$ is disentangled under $g$. By D1, each factor $f_i$ is obtained by pushing forward $p(z_i)$ along an axis-aligned line indirection $\boldsymbol{z}^i$, and the $i$-th seam follows $\boldsymbol{J}_z \boldsymbol{z}^i = \boldsymbol{J}_z^i$ at $z$ (column $i$ of $\boldsymbol{J}_z$). By factor independence (D1), only $f_i$ can change along the $i$-th seam, hence all other factors must be orthogonal to $\boldsymbol{J}_i$, i.e. $\boldsymbol{J}_z^\top \boldsymbol{J}_z$ is diagonal or $\boldsymbol{J}_z = \boldsymbol{U}_z \boldsymbol{S}_z$ (**C1**). Since $f_i$ depends on $s^i \doteq [\boldsymbol{S}_z]_{ii}$ and only $s^i$ can change along seam $i$, then $\frac{\partial s_i}{\partial z_j} = 0$, $i \neq j$ (**C2**). $\qquad\square$

## D.2   PROOF OF LINEAR VAE IDENTIFIABILITY

**Corollary 6.1** (**LVAE Identifiability**). *Let data be generated under the linear Gaussian LVM Eq. 2 with ground-truth $g(z) = \boldsymbol{W} z$, $\boldsymbol{W} = \boldsymbol{U_W} \boldsymbol{S_W} \boldsymbol{V_W}^\top \in \mathbb{R}^{m \times d}$ of full column rank and* distinct *singular values. Let an LVAE with* diagonal *posteriors be trained on $n$ samples, and as $n \to \infty$ its learned parameters yield $p_\mu^{(d)} \equiv p_\mu^{(g)}$ on the mean manifold. Then the LVAE achieves disentanglement (D1) and identifies ground-truth independent components on $\mathcal{M}_g$ up to permutation and sign (**P&S**).*

*Proof.* (**Ground truth**) With $\mu = \boldsymbol{W} z$ and $u \doteq \boldsymbol{U_W}^\top \mu = \boldsymbol{S_W}(\boldsymbol{V_W}^\top z)$, then $z \sim \mathcal{N}(0, \boldsymbol{I})$ and orthonormal $\boldsymbol{V_W}$ imply $\boldsymbol{V_W}^\top z \sim \mathcal{N}(0, \boldsymbol{I})$, hence $\{u_i \doteq u_{\boldsymbol{W},i}\}$ are independent, $u_i \sim \mathcal{N}(0, s_{\boldsymbol{W},i}^2)$ and $p_\mu^{(g)} = \prod_{i=1}^{d} \mathcal{N}(u_i; 0, s_{\boldsymbol{W},i}^2)$.

(**Model**) Let $d(z) = \boldsymbol{D} z$ with SVD $\boldsymbol{D} = \boldsymbol{U_D} \boldsymbol{S_D} \boldsymbol{V_D}^\top$. For an LVAE, the Hessian term in Eq. 6 is zero and Assumption 1 is trivially satisfied. Thus, by Lemma 4.1, right singular paths are axis–aligned (C1, note C2 is vacuous). Therefore, by the *disentanglement theorem* Theorem 5.2, $p_\mu^{(d)}$ is disentangled and factorises into statistically independent components along the decoder's seams (columns of $\boldsymbol{U_D}$). Since $u_{\boldsymbol{D},i} = s_{\boldsymbol{D},i} z_i$ and $z_i \sim \mathcal{N}(0, 1)$, each seam factor is Gaussian with variance $s_{\boldsymbol{D},i}^2$, i.e. $p_\mu^{(d)} = \prod_{i=1}^{d} \mathcal{N}(u_{\boldsymbol{D},i}; 0, s_{\boldsymbol{D},i}^2)$.

(**Matching**) Equality $p_\mu^{(d)} \equiv p_\mu^{(g)}$ and *distinct* $\{s_{\boldsymbol{W},i}\}$ imply uniqueness of the Gaussian product decomposition, up to permutation. Thus the LVAE's independent components (seam factors) match ground-truth components up to permutation/sign, i.e. identifiability and disentanglement on $\mathcal{M}_g$. $\quad\square$

## D.3   PROOF OF GAUSSIAN VAE IDENTIFIABILITY

**Lemma 6.3** (**Seams are Intrinsic**). *Let $\mathcal{M} \subseteq \mathcal{X}$ carry a manifold density $p_\mu$. Assume that on a regular set $\mathcal{M}_{\text{reg}}$ there exist scalar functions $\{u_i(x)\}_{i=1}^{d}$ (each varying only along a 1-D curve through $x$, i.e. a seam) and 1-D densities $\{f_i\}$ such that*

$$p_\mu(x) = \prod_{i=1}^{d} f_i(u_i(x)), \qquad x \in \mathcal{M}_{\text{reg}}, \tag{8}$$

*and that the on-manifold Hessian $\mathbf{H}_x \doteq \nabla_x^2 \log p_\mu(x)$ has pairwise distinct eigenvalues a.e. on $\mathcal{M}_{\mathrm{reg}}$. Then for each $x \in \mathcal{M}_{\mathrm{reg}}$, the $d$ seam directions (along which exactly one $u_i$ varies) are determined intrinsically by $p_\mu$, as eigenvectors of $\mathbf{H}_x$, unique up to permutation and sign (P&S).*

*Proof.* For $x \in \mathcal{M}_{\mathrm{reg}}$, let $\boldsymbol{u}^i, \dots, \boldsymbol{u}^d$ be unit tangent directions at $x$ such that, along $\boldsymbol{u}^i$, only $u_i$ varies locally while $u_{j \neq i}$ remain constant, thus $\{\boldsymbol{u}^i\}$ are orthonormal. Stack $\{\boldsymbol{u}^i\}$ as columns of $\boldsymbol{U}_x \in \mathbb{R}^{m \times d}$ and define seam coordinates $u(x) \doteq (u_1(x), \dots, u_d(x))$. Thus, for each $i$, $\frac{\partial x}{\partial u_i} = \boldsymbol{u}^i$ and by the chain rule,

$$[\nabla_u \log p_\mu(x)]_i = \tfrac{\partial}{\partial u_i} \log p_\mu(x) = (\tfrac{\partial x}{\partial u_i})^\top \nabla_x \log p_\mu(x) = \boldsymbol{u}^{i\top} \nabla_x \log p_\mu(x).$$

$$[\nabla_u^2 \log p_\mu(x)]_{ij} = (\tfrac{\partial^2 x}{\partial u_j \, \partial u_i})^\top \nabla_x \log p_\mu(x) + (\tfrac{\partial x}{\partial u_i})^\top \underbrace{[\nabla_x^2 \log p_\mu(x)]}_{\mathbf{H}_x} (\tfrac{\partial x}{\partial u_j}) = \boldsymbol{u}^{i\top} \mathbf{H}_x \, \boldsymbol{u}^j,$$

where the last equality uses that the basis vectors $\{\boldsymbol{u}^i(x)\}$ are fixed, so $\frac{\partial^2 x}{\partial u_j \, \partial u_i}(x) = 0$. In summary,

$$\nabla_u \log p_\mu(x) = \boldsymbol{U}_x^\top \nabla_x \log p_\mu(x), \qquad \nabla_u^2 \log p_\mu(x) = \boldsymbol{U}_x^\top \mathbf{H}_x \, \boldsymbol{U}_x,$$

$$\text{i.e.} \qquad \mathbf{H}_x = \boldsymbol{U}_x \, [\nabla_u^2 \log p_\mu(x)] \, \boldsymbol{U}_x^\top. \tag{15}$$

By Eq. 8, $\log p_\mu(x) = \sum_{i=1}^d \log f_i(u_i(x))$, hence the central term $\nabla_u^2 \log p_\mu(x)$ has components

$$\left[\nabla_u^2 \log p_\mu(x)\right]_{ij} = \begin{cases} \frac{\partial^2}{\partial u_i^2} \log f_i(u_i(x)) & (i = j), \\ 0 & (i \neq j), \end{cases}$$

and is *diagonal*. Thus Eq. 15 is an eigendecomposition with distinct eigenvalues (by assumption); and seam directions $\boldsymbol{u}^i$ are eigenvectors of $\nabla_u^2 \log p_\mu(x)$ and so are unique up to P&S. $\qquad \square$

**Implication for identifiability.** Lemma 6.3 isolates the intrinsic geometry of $p_\mu$: once $p_\mu$ is fixed, the seams and their directions are fixed (P&S). Any Gaussian VAE decoder $d$ matching $p_\mu$ and satisfying **C1–C2** must therefore align its singular paths with those seams and inherit the same seam factors.

**Lemma 6.4 (A matching decoder finds seams).** *Under assumptions of Lemma 6.3, let $d : \mathcal{Z} \to \mathcal{X}$ be* c.i.d.a.e. *with factorised prior $p(z) = \prod_i p_i(z_i)$ and push-forward density $p_\mu^{(d)} \equiv p_\mu$ matching on $\mathcal{M}_d \doteq \{d(z)\} = \mathcal{M}$. If $d$ satisfies **C1–C2** a.e., then for any $z$ and $x = d(z)$:*

- *left singular vectors $\boldsymbol{U}_z$ of $\boldsymbol{J}_z$ coincide with the seam directions in Lemma 6.3 (up to P&S);*
- *the images under $d$ of singular-vector paths (D3) are exactly the seams through $x$;*
- *along the $i$-th seam, the factor $f_i$ is the 1-D push-forward of $p_i(z_i)$ (as in Lemma 5.1).*

*Proof.* Since $p_\mu^{(d)} \equiv p_\mu$, both induce the same $\mathbf{H}_x = \nabla_x^2 \log p_\mu(x)$. By Lemma 5.1 and **C1**, $\log p_\mu(x) = \sum_{i=1}^d \big( \log p_i(z_i) - \log s_i(z) \big)$, with $s_i$ depending only on $z_i$ by **C2**. Letting $u = \boldsymbol{U}_z^\top x$, the gradient along the manifold (*on-manifold score*) is given by

$$\nabla_x \log p_\mu(x) = \boldsymbol{U}_z \nabla_u \log p_\mu(x), \qquad \left[\nabla_u \log p_\mu(x)\right]_i = \tfrac{1}{s_i(z)} \tfrac{\partial}{\partial z_i}\big( \log p_i(z_i) - \log s_i(z) \big).$$

Differentiating again along the manifold gives the *on-manifold Hessian* $\mathbf{H}_x$

$$\nabla_x^2 \log p_\mu(x) = \boldsymbol{U}_z \, [\nabla_u^2 \log p_\mu(x)] \, \boldsymbol{U}_z^\top, \qquad [\nabla_u^2 \log p_\mu(x)]_{ij} = \begin{cases} \frac{1}{s_i} \frac{\partial}{\partial z_i} [\nabla_u \log p_\mu(x)]_i & (i = j) \\ 0 & (i \neq j) \end{cases}$$

Hence,

$$\mathbf{H}_x = \boldsymbol{U}_z \, \mathrm{Diag}\big( \tfrac{1}{s_i(z)} \tfrac{\partial}{\partial z_i} \big[\nabla_u \log p_\mu(x)\big]_i \big) \, \boldsymbol{U}_z^\top$$

is an eigendecomposition. With a simple spectrum, eigenvectors are unique up to P&S and coincide with the intrinsic seam directions in Lemma 6.3. By Lemma C.1, singular–vector paths map to seams; and by Lemma 5.1, the factor along seam $i$ equals the 1-D push-forward of $p_i(z_i)$. $\qquad \square$

Note that the two proofs above adopt a similar technique, but Lemma 6.3 is entirely intrinsic to the manifold (hence no mention of a Jacobian), whereas Lemma 6.4 is with reference to a parameterisation of the manifold by a function $d$.

### D.4 PROOF OF GAUSSIAN VAE IDENTIFIABILITY

**Corollary D.1** (**Gaussian VAE Identifiability**). *Let data be generated by* c.i.d.a.e. $g : \mathcal{Z} \to \mathcal{X}$ *with factorised prior* $p(z) = \prod_{i=1}^{d} p_i(z_i)$*; let a Gaussian VAE with* diagonal *posteriors learn a decoder* $d : \mathcal{Z} \to \mathcal{X}$*. Suppose both g and d satisfy **C1–C2** a.e. and manifold densities match,* $p_\mu^{(d)} \equiv p_\mu^{(g)}$*, on the common manifold* $\mathcal{M} = \{g(z)\} = \{d(z)\}$*. If the tangent Hessian* $\mathbf{H}_x$ *has a simple spectrum a.e., then d identifies the ground-truth seam decomposition (independent components) of* $p_\mu^{(g)}$*, up to P&S.*

*Proof.* (**Matching** $U$) Equality $p_\mu^{(g)} \equiv p_\mu^{(d)}$ implies the same $\mathbf{H}_x$. By Lemma 6.3, its eigenvectors are intrinsic and unique (P&S), so $\boldsymbol{U}_z^{(d)} = \boldsymbol{U}_z^{(g)}$ (P&S, herein assume indices are relabelled to match).

(**Matching** $S$) In this common basis, under **C1–C2**, on-manifold scores are equal:

$$\left[ \nabla_u \log p_\mu^{(g)}(x) \right]_i \;=\; \frac{1}{s_i^{(g)}(z)} \frac{\partial}{\partial z_i} \Big( \log p_i(z_i) - \log s_i^{(g)}(z) \Big) \;=\; \left[ \nabla_u \log p_\mu^{(d)}(x) \right]_i,$$

hence integrating along seam $i$ (with $z_{\neg i}$ fixed, using equality of $p_\mu$ at a reference point to fix the integration constant), 1-D seam factors $\frac{p_i(z_i)}{s_i}$ match. Since $p_i$ is fixed, $s_i^{(d)} = s_i^{(g)}$, i.e. $\boldsymbol{S}_z^{(d)} = \boldsymbol{S}_z^{(g)}$.

With $\boldsymbol{V}_z^{(d)} = \boldsymbol{V}_z^{(g)} = \boldsymbol{I}$ by **C1**, it follows that $\boldsymbol{J}_z^{(d)} = \boldsymbol{J}_z^{(g)}$ and the seam decomposition is identified up to P&S. $\square$

## E DISENTANGLEMENT METRICS

**Axis alignment score (AAS)**: Given a matrix of mutual information values, between each latent co-ordinate and each ground truth factor, one can normalise over rows or columns to compute a "distribution" of mutual information.

The entropy of each distribution gives a measure of how narrowly or sparsely information about a ground truth factor is captured across latents or the spread of information about each factor captured by a single latent. In either case, a "high entropy" distribution means information is widely spread, while low entropy means information about a factor is concentrated in a single latent, i.e. disentangled.

Entropy of the mutual information distribution can be computed row-wise or column-wise. AAS is a holistic metric combining the intuitions of both options into a single, robust score that evaluates how close matrix M is to a permuted diagonal form (zero entropy, perfect disentanglement).

In a perfectly disentangled MI matrix, the sum of peak values per row equals the sum of peak values per column, and both equal the total sum of the matrix. AAS measures the ratio of the "sum of peaks" to the "total sum":

```
sum_col_max = sum(max(mut_info, dim=0))
sum_row_max = sum(max(mut_info, dim=1)
aas = 0.5 * (sum_row_max + sum_col_max) / sum(mut_info)
```

**Normalised off diagonal**: For gradient terms (here, a $d \times d$ matrix $M$) we compute a measure of diagonality by computing the ratios of normalised off-diagonal absolute values to on-diagonal values.

```
d = M.shape[1]
num_off_diag = m * (m - 1)
M = abs(M)
M = diag(M)^(-0.5) * M * diag(M)^(-0.5)       # normalise
mean_off_diag = (sum(M) - sum(diag(M))) / num_off_diag
```

# F EMPIRICAL RESULTS ON NATURAL DATA (CELEBA)

This appendix complements §7, applying the same architecture and training regime used for dSprites (and as in Burgess et al. (2018)) to CelebA, a more complex natural dataset without introducing new confounds. We vary $\beta$ and compare *diagonal* vs *full* posterior covariances, reporting: (i) latent traversals showing the dependence of disentanglement on posterior structure (Fig. 7); (ii) how utilisation of the latent space varies with $\beta$ and posterior structure (Fig. 8); (iii) diagonality of the Price/Bonnet derivative terms (Fig. 9); and (iv) reconstruction/sampling quality (10). See captions for details.

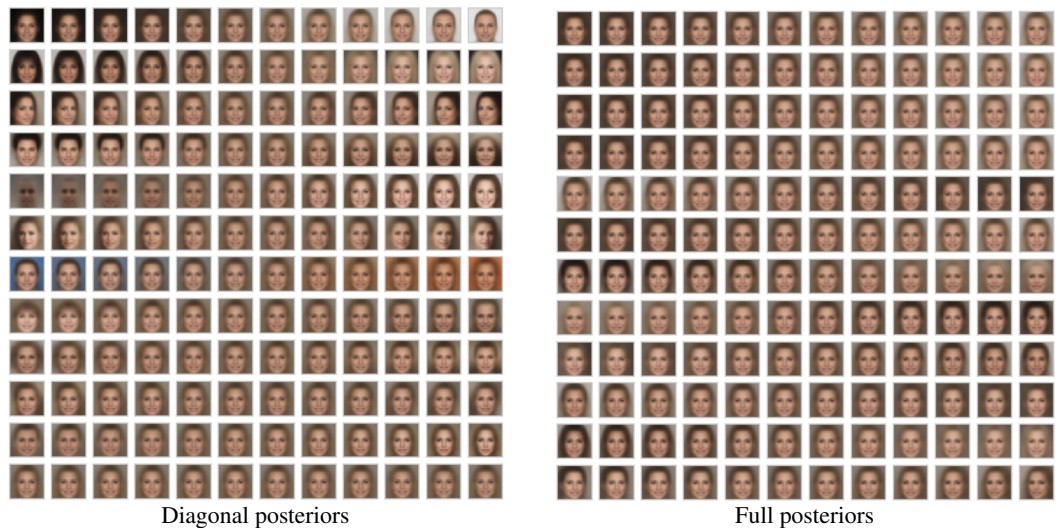

|          Diagonal posteriors          |          Full posteriors          |

Figure 7: **Traversals over dimensions of highest variance** ($\beta = 4$): Each row shows images generated as individual latent dimensions $z_i$ are varied with rows ordered by latent activity (See Fig. 8 caption). For diagonal posteriors (left), traversals more clearly demonstrate disentanglement, i.e. identification of distinct semantic features with distinct latent dimensions, e.g. background shade (row 1), facial orientation (row 3), lighting (row 5), background colour (row 7). For full covariance posteriors, independent features are less clearly assigned to distinct latent dimensions, i.e. less disentangled.

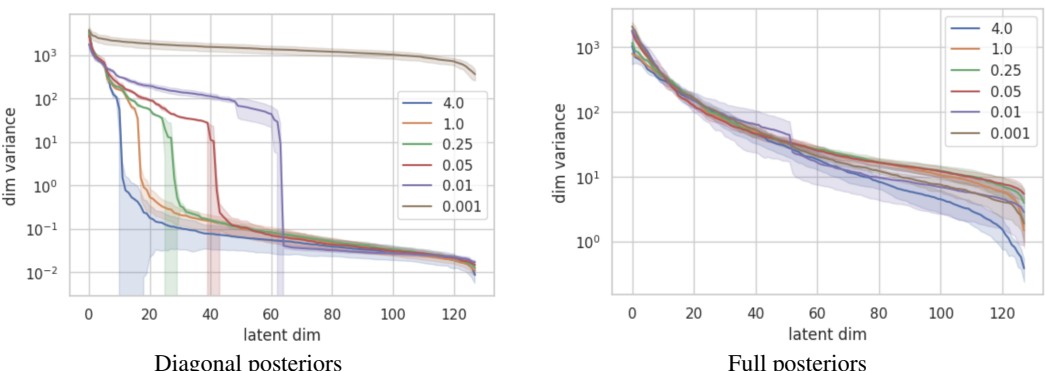

|          Diagonal posteriors          |          Full posteriors          |

Figure 8: **Active latent dimension depend on $\beta$ (denoted by colour) and posterior covariance structure (left/right)**: For a trained model and for each latent dimension $z_i$, the variance $x|z_i$ is estimated by taking equidistant traversals in latent space and computing the Euclidean distance between samples at each end of the traversal. The plot show the (estimated) variance, or *latent activity*, per dimension ordered by magnitude (log scale, mean over 5 runs, standard deviation indicated by shaded areas). With diagonal covariances (left), there are relatively sharp cliff-edges implying that dimensions are (broadly) *active* or *inactive* and that axis-aligned directions are preferred. The number of active dimension increases as $\beta$ reduces and reconstructions get "sharper", requiring more information to be captured and thus greater capacity (i.e. latent dimensions). For full posteriors (right), the distribution of variance over dimensions is much smoother and axis-aligned directions have no special status.

$\beta$

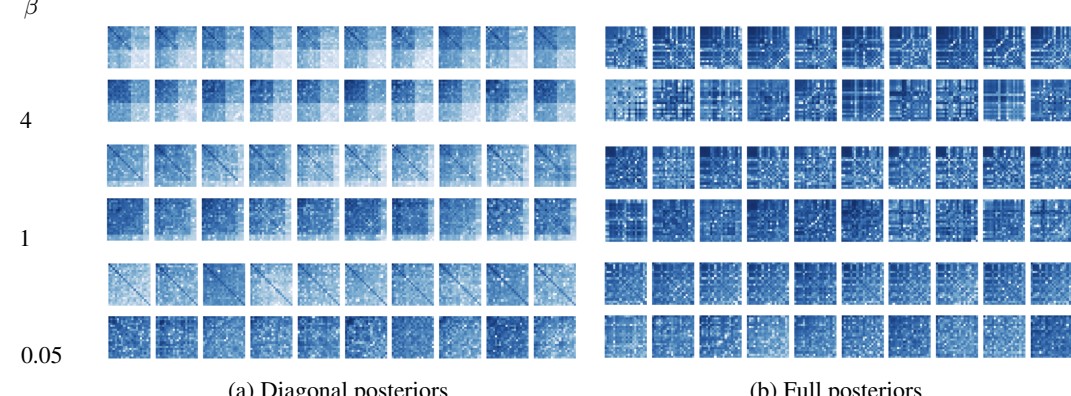

(a) Diagonal posteriors                    (b) Full posteriors

Figure 9: **Heatmaps of Derivative Terms in Eq. 6 for CelebA for Diagonal and Full Posterior Covariances**: For each value of $\beta$ (indicated on left), there are two rows for $\boldsymbol{J}^\top \boldsymbol{J}$ (upper) and the directed Hessian (lower) over 10 random test samples. Colour intensity indicates log magnitude of matrix entries. For each heatmap, rows and columns correspond to a latent dimensions $z_i$, ordered by latent activity (See Fig. 8 caption). We show the top 20 most active dimensions. For diagonal covariances (left), the active dimensions are visible as a darker block in the upper left, which grows with as $\beta$ reduces (matching Fig. 8), and diagonal structure is visible for active dimensions of $\boldsymbol{J}^\top \boldsymbol{J}$. Such structure is not visible for full covariances (right). The Hessians show less discernable structure and we suspect that such a higher order derivative require more samples to be well estimated for a complex distribution.

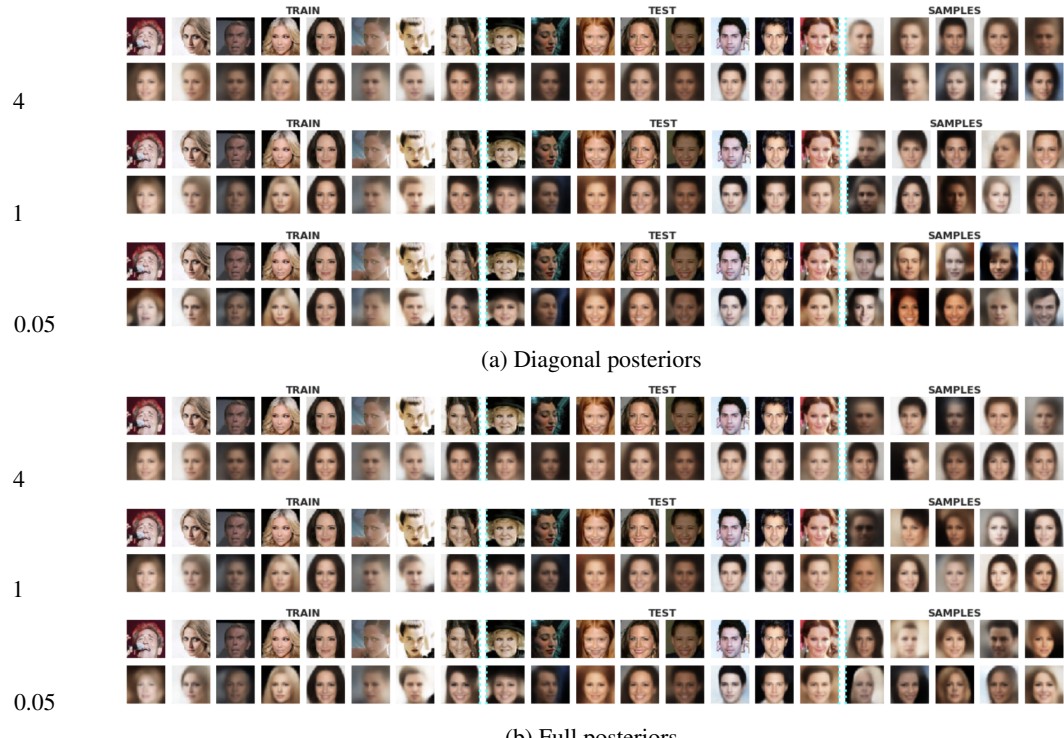

(a) Diagonal posteriors

(b) Full posteriors

Figure 10: **Reconstructions and Samples for CelebA for a range of $\beta$ values and Diagonal and Full Posterior Covariances**: For each value of $\beta$ (left), there are two rows in three sections: (left) train samples (upper) and reconstructions (lower); (mid) test samples (upper) and reconstructions (lower); and (right) samples (both rows). As $\beta$ reduces, reconstruction quality and samples improve (i.e. blur reduces). Reconstruction and sample quality is broadly comparable for diagonal and full covariances, indicating that the latent space is reoriented towards axis-alignment without necessarily impacting performance.

**Summary:** For *diagonal* posteriors, we observe: (a) a small set of *active* latents whose number increases as $\beta$ decreases; (b) stronger Jacobian orthogonality among active dimensions; and (c) reconstructions/samples of comparable quality to the full-covariance model across $\beta$ (Figures 7–10). These match the predictions of our theory and mirror the synthetic/dSprites trends, supporting on natural data the claim that diagonal posteriors drive C1–C2 in expectation.

# G REDUCING $\beta$ OVER TRAINING

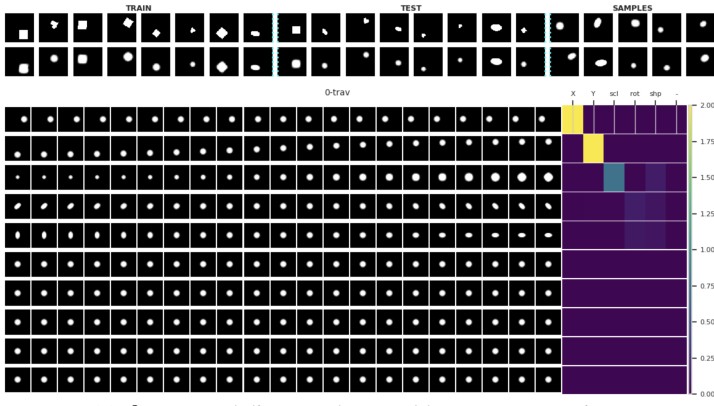

(a) $\beta = 1$: good disentanglement, blurry reconstructions.

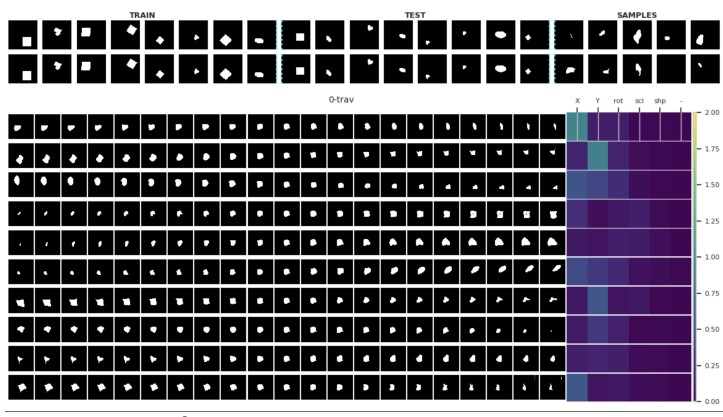

(b) $\beta = 10^{-3}$: no clear disentanglement, good reconstructions.

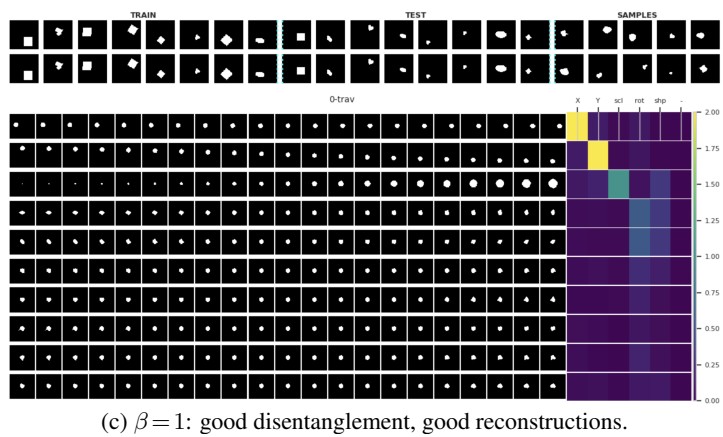

(c) $\beta = 1$: good disentanglement, good reconstructions.

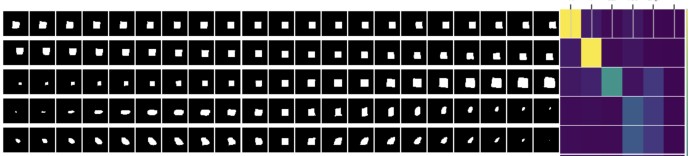

(d) Traversals from a random test sample

Figure 11: **Testing the $\beta$-hypothesis:** (top) high $\beta$ (1) gives best disentanglement (see heatmap) but blurry images (see top rows); (mid) low $\beta$ (0.001) gives poor disentanglement but good reconstructions; (bottom) lowering $\beta$ over training ($1 \rightarrow 0.001$) gives good disentanglement (see heatmap) and good reconstructions.

## H  MATERIAL ERRORS IN REIZINGER ET AL. (2022)

We note what appear to be several fundamental mathematical errors in the proof of Theorem 1 in Reizinger et al. (2022) rendering it invalid. Theorem 1 claims an approximation to the exact relationship given in Eq. 6

1. p.33, after "triangle inequality": $\left|\mathbb{E}\left[\|a\|^2 - \|b\|^2\right]\right| \leq \mathbb{E}\left[\|a-b\|^2\right]$, where $a = x-f$, $b = -\sum \frac{\partial f}{\partial z_k}...$

   - (dropping expectations for clarity) this has the form $\left|\|a\|^2 - \|b\|^2\right| \leq \|a-b\|^2$  (*)
   - true triangle inequality:  $\left|\|a\|-\|b\|\right| \leq \|a-b\| \implies \left|\|a\|-\|b\|\right|^2 \leq \|a-b\|^2$ (by squaring)
     - this differs to (*) since norms are squared inside the absolute operator on the L.H.S.
   - counter-example to (*):  $b = x > 0$, $a = x+1 \implies \left|\|a\|^2 - \|b\|^2\right| = |2x+1| > 1 = \|a-b\|^2$

2. next step, p.33:  $\mathbb{E}\left[\|(c-e) - (d-e)\|^2\right] \leq \mathbb{E}\left[\|c-e\|^2 + \|d-e\|^2\right]$  where $c = x$, $d = f(z) - \sum \frac{\partial f}{\partial z_k}...$, $e = f(\mu)$

   - this has the form of the standard triangle inequality $\|a-b\| \leq \|a\|+\|b\|$ except all norms are squared.
   - squaring both sides of the triangle inequality gives an additional cross term on the right that the used inequality omits, without which the inequality does not hold in general.

3. first step, p.34: drops the K term, which bounds the decoder Hessian and higher derivatives (in earlier Taylor expansion)

   - this omission is similar to a step in Kumar & Poole (2020) but is not stated, e.g. in Assumption 1.
   - since K is unbounded, any conclusion omitting it without justification is not valid in general.

## I  THE USE OF LARGE LANGUAGE MODELS (LLMs)

LLMs were used to assist drafting this paper as follows:

- general review for errors, inconsistencies and readability;
- verifying proofs, generating code snippets or identifying code errors;
- creating figure 1.

