# OpenReview forum: "Unpicking Data at the Seams: Understanding Disentanglement in VAEs"
_ICLR.cc/2026/Conference — Submitted to ICLR 2026_

### Official Review · Reviewer_8oT4 · 2025-10-27

**Soundness:** 3
**Presentation:** 3
**Contribution:** 3
**Rating:** 6
**Confidence:** 4

**Summary:**

This paper provides a theoretical framework for understanding disentanglement in Variational Autoencoders (VAEs), explaining why and how VAEs learn to separate independent generative factors of data without explicit supervision.
The authors define disentanglement as factorizing the density over a data manifold into independent one-dimensional "seam" factors, where each factor is the push-forward of density over an axis-aligned latent path.
They prove that 2 conditions on the decoder is equivalent to disentanglement.
They also prove the identifiability of the LVAE and Gaussian VAE models.
Experiments on dSprites dataset confirm that diagonal posterior covariances promote diagonalized derivative terms and that diagonality correlates with disentanglement metrics. The paper shows diagonal covariance VAEs achieve better disentanglement scores (MIG and proposed AAS metric) compared to full covariance VAEs, with individual latent coordinates correlating with distinct ground truth factors.

**Strengths:**

All the claims are proved theoretically providing solid bases for understanding this disentanglement phenomenon.

Correction of a previous paper is useful.

**Weaknesses:**

Very low experimental support is provided.
It would have been interesting to have an idea of the required dimension of the latent space to reach the whole disentanglement. Can it be related to the noisy ICA decomposition and the number of independent factors?
What is the effect of on reconstruction when the C1-C2 conditions are satisfied compared to other VAE-like models?
What is the effect on generation as well?
Is this property useful when trying to use VAE with a small dataset?

**Questions:**

See weaknesses.

---

> ### Author Response · Authors · 2025-11-21
>
> Many thanks for your review and your recognition of our work as a "solid basis" for understanding the well-known phenomenon of disentanglement, the key aim of our paper.
>
> **Summary**:
> * The key purpose of our paper is a theoretical understanding of disentanglement.
> * Our work extends prior published works by providing a rigorous distributional definition of disentanglement, precise necessary-and-sufficient conditions on the decoder, and identifiability results for the induced factors.
> * Given the wealth of previous empirical studies, we include targeted experiments directly connected to our theory.
> * We have added **CelebA** analysis (App. G) to demonstrate these properties on a natural, more complex dataset.
> * Our analysis offers a principled basis for future work and experimental design.
>
> **Response to points/questions**:
>
>
> 1. **Low experimental support**:
>
>    * Our core objective is a **rigorous mathematical understanding of disentanglement** and its relationship to data structure. A succession of published works have aimed to explain it, which we build on to:
>      * give a **rigorous definition of disentanglement**;
>      * **characterise precise decoder conditions** for it (independent of training regime);
>      * show that diagonal covariances induce such conditions (in expectation) and how $\\beta$ fits within our framework, aligning with empirical observations; and
>      * **prove that independent factors are identifiable** up to natural symmetry.
>
>    * Since many prior works demonstrate disentanglement and Jacobian orthogonality on a variety of synthetic and natural datasets, we perform targeted experiments to illustrate the theory:
>      * experiments on synthetic Gaussian data (analytic, trackable ground truth) and dSprites (known factors) allow precise correlation calculations between model and ground-truth factors.
>      * Fig 2 shows **Gaussian prior symmetry breaking** by diagonal covariances and **independent component identifiability**, even where classical noisy‑ICA is *unidentifiable*;
>      * Fig. 5 shows **directed Hessian diagonality** (further to known Jacobian orthogonality);
>      * our analysis suggests that $\\beta$ trades off output noise and disentanglement hence reducing it (a known heuristic) may give the best of both. While not the subject of this paper, we demonstrate this, showing it deserves future study;
>      * given your feedback **we have included CelebA analysis** (Appendix F), which shows
>         * "disentanglement" on a natural, more complex dataset (Fig. 8);
>         * that for diagonal covariances (only), latent dimensions tend to be "active"/"inactive" (i.e. information is axis aligned in latent space); and the number of active latents increases as $\\beta$ reduces since reconstructions become sharper requiring more information to be captured (Fig. 9);
>         * that diagonal covariances promote diagonality of the Jacobian term in active dimensions (Fig. 10);
>         * that reconstruction and sample quality are comparable for diagonal vs full covariances (Fig 11).
>
> 2. **Required latent dimension for disentanglement**:
>    * to match the target distribution via pushforward, **latent dimensionality should match the number of ground truth factors**.
>    * diagonal posteriors align factors with latent dimensions so redundant factors revert to the prior (see "active"/"inactive" dims, Fig. 9).
>    * With too few latents, the loss is minimized by learning factors of greatest variance (as in PCA).
>
> 3. **Relation to noisy ICA and number of factors**:
>       * **Noisy ICA [[1]](#noisyICA) and a Gaussian VAE assume the same generative model but their loss function differs**.
>       * ICA does not model the posterior $q(z|x)$, which promotes disentanglement in a VAE (see Remark 6.8).
>       * We have added reference to noisy ICA in Related Works.
>       * Both models can only fit the data distribution if their latent dimension matches the number of ground truth independent factors.
>
> 4. **Effect on reconstruction and generationof C1-C2**:
>      * C1–C2 are **induced by the ELBO with diagonal posteriors** (in expectation) and are induced in any VAE with diagonal posteriors (the default).
>     * when ground truth factors are independent, C1-C2 **select among equivalent optima** (as in the linear case, Fig. 2);  when independence is imperfect, they encourage diagonality (dSprites, Fig 5; CelebA, Fig 10).
>     * on CelebA, diagonal vs full posteriors yield comparable reconstructions and samples (Fig. 11).
>
> To reiterate, our work **extends prior published works** on disentanglement and offers insight into generative model structure more general than VAEs. We thank you again for your time and hope that the CelebA analysis and above clarifications address your empirical‑scope concerns while keeping the paper focused on its core theoretical contribution.
>
>
> [1]  "Noisy independent component analysis, maximum likelihood estimation, and competitive learning", Hvarinen, 1998.

---

### Official Review · Reviewer_GTYe · 2025-11-01

**Soundness:** 3
**Presentation:** 3
**Contribution:** 3
**Rating:** 6
**Confidence:** 3

**Summary:**

This paper provides a theoretical explanation for why Variational Autoencoders (VAEs) often exhibit latent disentanglement—the phenomenon where manipulating a single latent dimension changes only one semantic factor in the generated data (e.g., object position, facial expression). The authors argue that disentanglement in VAEs arises largely due to the common practice of using diagonal Gaussian posterior covariances, not due to explicit disentangling losses. They demonstrate an exact mathematical link between optimal Gaussian posteriors and decoder Jacobians, showing that diagonal posterior assumptions “lock” the decoder into axis-aligned directions in latent space. As a result, the learned data density factorizes into independent 1-D seams corresponding to latent dimensions—producing disentanglement “for free.”

**Strengths:**

1. Novelty. The paper formalizes this mechanism and proves conditions under which latent factors are identifiable fro VAE, even with a symmetric prior, addressing the long-standing “unidentifiability” concern of nonlinear latent models. It also clarifies how \beta-VAEs enhance this effect by modulating posterior variance and reducing posterior collapse.

2. Theoretical value: Theoretical proof connecting diagonal posterior covariances to axis-aligned latent factorization through VAE Jacobians. It provides explanation of how β-VAE (β > 1) improves disentanglement by controlling posterior variance.

3. Empirical demonstration shows that true generative factors are identifiable in VAEs under stated assumptions.

**Weaknesses:**

Limited Empirical Scope

1. The experiments primarily use: Synthetic linear-Gaussian settings, dSprites dataset (simple factors, low resolution), Small-scale VAEs (dim=10)

2. While appropriate for theory illustration, they do not adequately test: complex natural image datasets (such as CelebA, Imagenet1K)
Empirical evidence also lacks real-world factor disentanglement settings (e.g., pose, lighting, angle)

3. Claims about generality such as Diffusion or GAN architectures discussed as motivation are not yet experimentally supported, although this is minor.

**Questions:**

Please see weakness for questions.

---

> ### Author Response · Authors · 2025-11-21
>
> Many thanks for your review, positive comments and recognising that our work addresses the long-standing phenomenon of disentanglement and identifiability, the key aim of our paper.
>
>
> **Summary**:
> * The key purpose of our paper is a theoretical understanding of disentanglement.
> * Our work extends prior published works by providing a rigorous distributional definition of disentanglement, precise necessary-and-sufficient conditions on the decoder, and identifiability results for the induced factors.
> * Given the wealth of previous empirical studies, we include targeted experiments directly connected to our theory.
> * We have added **CelebA** analysis (App. G) to demonstrate these properties on a natural, more complex dataset.
> * Our analysis offers a principled basis for future work and experimental design.
>
>
> 1. **Empirical Scope (synthetic, small scale, not complex natural datasets)**:
>    * Our core objective is a **rigorous mathematical understanding of disentanglement** and its relationship to data structure. A succession of published works have aimed to explain it, which we build on to:
>      * give a **rigorous distributional definition of disentanglement**;
>      * **characterise precise decoder conditions** for it (independent of training regime);
>      * show that diagonal covariances induce such conditions (in expectation) and how $\\beta$ fits within our framework, aligning with empirical observations; and
>      * **prove that independent factors are identifiable** up to natural symmetry.
>
>    * Since many prior works demonstrate disentanglement and Jacobian orthogonality on a variety of synthetic and natural datasets, we perform targeted experiments to illustrate the theory:
>      * experiments on synthetic Gaussian data (analytic, trackable ground truth) and dSprites (known factors) allow precise correlation calculations between model and ground-truth factors.
>      * Fig 2 shows **Gaussian prior symmetry breaking** by diagonal covariances and **independent component identifiability**, even where classical noisy‑ICA is *unidentifiable*;
>      * Fig. 5 shows **directed Hessian diagonality** (further to known Jacobian orthogonality);
>      * our analysis suggests that $\\beta$ trades off output noise and disentanglement hence reducing it (a known heuristic) may give the best of both. While not the subject of this paper, we demonstrate this, showing it deserves future study;
>      * per the reviewer's suggestion **we include CelebA analysis** (Appendix F), which shows
>         * "disentanglement" on a natural, more complex dataset (Fig. 8);
>         * that for diagonal covariances (only), latent dimensions tend to be "active"/"inactive" (i.e. information is axis aligned in latent space); and the number of active latents increases as $\\beta$ reduces since reconstructions become sharper requiring more information to be captured (Fig. 9);
>         * that diagonal covariances promote diagonality of the Jacobian term in active dimensions (Fig. 10);
>         * that reconstruction and sample quality are comparable for diagonal vs full covariances (Fig 11).
>
> 2. **Generality**:
>     * While we study “disentanglement in VAEs,” our analysis is formulated at the level of pushforward densities; the key results (Lem. 5.1; Thm. 5.2) are model‑agnostic under their stated smoothness/regularity assumptions and thus apply at the pushforward level to decoders/generators beyond VAEs.
>     * We do not claim that other objectives (e.g. GANs, diffusion) empirically induce the same conditions; establishing when those training regimes enforce analogous structure is beyond our current scope but a natural direction for future work.
>     * The contribution we look to emphasise to the community is that several of our most significant results are not specific to VAEs; VAEs with diagonal posteriors are one mechanism that induces the conditions we identify (C1–C2).
>     * **We have made this more clear in the paper**
>
> To reiterate, we believe our work **extends prior published works** on disentanglement and offers insight more general than VAEs. We thank you again for your time and hope that the CelebA analysis and clarifications above address your empirical‑scope concerns while keeping the paper focused on its core theoretical contribution.

---

### Official Review · Reviewer_iHbB · 2025-11-02

**Soundness:** 1
**Presentation:** 2
**Contribution:** 2
**Rating:** 2
**Confidence:** 2

**Summary:**

This paper provides a precise theoretical explanation for why Variational Autoencoders (VAEs) with diagonal posterior covariances tend to learn disentangled representations. Disentanglement is the phenomenon where varying a single latent coordinate causes a single, semantically meaningful change in the generated sample.

The authors build their argument on an exact relationship between the optimal Gaussian posterior and the decoder's derivatives. They show that forcing the posterior covariance to be diagonal imposes constraints on the decoder's Jacobian ($J_z$) and Hessian ($H_z$). These constraints, in turn, imply two key properties must hold (in expectation):

1. **(C1)** Right singular vectors Vz of the decoder Jacobian Jz are standard basis vectors for all z ∈ Z, i.e. after relabeling/sign flips of the latent axes, we have Vz = I;
2. **(C2)** The matrix of partial derivatives of singular values ( ∂si∂zj )i,j is diagonal, i.e. ∂si∂zj = 0 for all i = j.

**Strengths:**

1. Formal Definition of Disentanglement: The paper proposes a precise and geometrically intuitive definition of disentanglement (D1). By defining it as the factorization of the manifold density into independent 1-D "seams", it moves the concept from a vague empirical observation to a testable mathematical property.
2. Precise Theoretical Mechanism: It provides a full theoretical explanation  for a long-observed phenomenon. Instead of relying on approximations , it uses an exact relationship from the Price/Bonnet Theorem to build a rigorous, step-by-step argument connecting diagonal posteriors to the geometric constraints (C1, C2) that provably guarantee disentanglement.

**Weaknesses:**

1. The proof is limited in Linear situation. These conclusions may not be applied in complex data. The errors caused by non-linearity are not discussed.
2. The conclusions in the paper are vanilla that some papers have explained. For instance, beta controls the regularization strength for index-code mutual information, total correlation, and dimension-wise KL [1]. This can explain the importance of beta and posing low model variance.
3. The impact of this research is unclear. This work has no experiment to prove the benefits derived from these theories. Annealing beta is a common trick in VAEs and disentanglement learning.


[1] Chen, Ricky TQ, et al. "Isolating sources of disentanglement in variational autoencoders." Advances in neural information processing systems 31 (2018).

**Questions:**

Can the proposed theory guide the design of novel disentanglement methods?
What is the novelty of the proposed theory and the comparison to beta-TCVAE?

---

> ### Author Response · Authors · 2025-11-21
>
> Many thanks for your review, positive summary and recognising that our work addresses the long-standing phenomenon of disentanglement and identifiability, the key aim of our paper.
>
> **Summary**:
> * The key purpose of the paper is to present **a fuller theoretical understanding of disentanglement**, which, to our knowledge, extends prior works by making precise: **a rigorous distributional definition** (D1), **necessary‑and‑sufficient decoder conditions** (C1–C2; Thm 5.2 from Lem 5.1), and **identifiability** of factors (§6).
> * We connect these to VAEs via an exact identity (§4), explaining how diagonal posteriors push decoders toward disentanglement and how $\\beta$ modulates this effect - within that framework.
> * Given the many prior works, we include targeted experiments targeted to our theory.
> * Our analysis offers a principled basis for future work and experimental design.
> * We have added **CelebA** analysis (App. F) to demonstrate these properties on a natural, more complex dataset.
>
>
> Weaknesses
> 1. **Proof limited to linear**:
>     * This seems to be a misunderstanding: **all of our key results are non-linear**. Def. D1, Lem. 5.1, and Thm 5.2 are stated for general smooth decoders and pushforward densities; the linear setting is included for intuition and full analytic tractability.
>
> 2. **“Vanilla conclusions”**
>    Our main contributions are not the qualitative observation that $\\beta$ affects variance (which we do not claim as novel), but rather:
>    * a **rigorous definition** of disentanglement (D1);
>    * a proof of **necessary and sufficient conditions** (C1–C2) for disentanglement (Thm 5.2);
>    * a proof that the resulting factors are **identifiable up to natural symmetries** (§6); and
>    * an explanation, via an exact identity (§4), of **why** diagonal posteriors and $\\beta$ push the decoder toward those conditions.
>
>    β‑TCVAE proposes a **loss** to encourage factorised latents, whereas our theory **characterises when the learned manifold density factorises** and establishes **identifiability** under stated assumptions. The two lines are orthogonal and complementary.
>
> 3. **Impact  & (Q1)**:
>    The paper is **primarily theoretical**. Our experiments are **targeted**: they measure the Price/Bonnet **Jacobian + directed‑Hessian** terms, show their **diagonality** increases with $\\beta$, and relate this to disentanglement metrics, alongside linear‑case identifiability. We believe our work provides a rigorous foundation for future research, e.g. suggesting that the known $\\beta$-annealing heuristic may have a principled justification that warrants further study. **We have made this more clear in the paper.**
>
> 4. **(Q2) Novelty vs $\\beta$-TCVAE**:
> $\\beta$-TCVAE decomposes the ELBO and **penalises total correlation of the aggregated posterior**, encouraging axis‑aligned latents. Our work is **orthogonal**: it (i) defines disentanglement as **manifold‑density factorisation** (D1), (ii) provides **if and only if** decoder conditions (C1–C2, Thm 5.2), (iii) proves **identifiability** (§6), and (iv) explains why diagonal posteriors and $\\beta$ induce those conditions in VAEs.
>    * In short: **TCVAE specifies a loss focusing on manipulating the prior; our theory characterises the geometry and identifiability of the resulting solutions**.
>
> To reiterate, our work is **non‑linear** in scope and provides novel results (***iff*** characterisations, **identifiability**, explaining the role of diagonal posteriors and $\\beta$) through an exact analysis. We appreciate the reviewer’s comments and hope that these clarifications address your concerns.

---

> > ### Comment · Reviewer_iHbB · 2025-11-28
> >
> > Thank you for the detailed response.
> >
> > For a short comparison, betaVAE penalizes encoders while your constraint apply on decoders. Is it right?

---

> > > ### Author Response · Authors · 2025-11-29
> > >
> > > Thanks for following up.
> > >
> > > It is not so easy to separate since $\\beta$‑TCVAE penalises the aggregate posterior, which is both the encoder output and the decoder input, and so affects both networks. Instead one could perhaps say:
> > > * $\\beta$‑TCVAE adds a constraint to enhance the disentanglement known to occur in $\\beta$-VAEs with diagonal posteriors.
> > > * We explain from first principles the source of disentanglement in $\\beta$-VAEs with diagonal posteriors: defining disentanglement in terms of statistical independence (D1) and showing how a VAE with diagonal posteriors induces it.
> > >
> > > We hope this resolves any remaining concerns beyond our previous clarification that none of our results are linear and our theoretical results are novel, particularly given the known *unidentifiability* results from ICA (discussed in Remark 6.8).

---

### Official Review · Reviewer_GVNd · 2025-11-03

**Soundness:** 3
**Presentation:** 2
**Contribution:** 2
**Rating:** 2
**Confidence:** 3

**Summary:**

This paper aims to develop a theoretical understanding of VAEs and their capacity to perform disentanglement. To do this, it builds upon prior theoretical results to argue that VAEs induce a pushforward distribution of the prior under a decoder which factorizes along the pushforward of each latent component. The authors then argue theoretically that this property yields identifiability. An experimental study is conducted which aims to confirm these theoretical claims empirically.

**Strengths:**

* I found the theoretical sections to be very well written and logically structured making the paper straightforward to digest and pleasant to read.


* Lemmas 4.1 and 5.1 as well as the theoretical results in Section 6 are potentially interesting for the identifiability and VAE communities.


* The issue regarding the proof in Reizinger et. al, 2022 is an interesting and important observation.


* All figures in the paper are very nicely done and informative.

**Weaknesses:**

The paper aims to make a contribution to both VAE theory and identifiability theory by analyzing the structure of a VAE decoder's derivative matrices. These are well studied research areas with a long line of theoretical results on understanding VAEs [1, 2, 3, 4, 5], particularly regarding their Jacobian and Hessian structure, as well as on identifiability (see [6] for a review), particularly for functions with orthogonal Jacobians [7,8,9] and diagonal Hessians [10, 11].


This is by no means to say these areas are saturated. However, I believe it is very important for any work in this space to position itself very clearly relative to prior works, in order to understand precisely what is novel. *In my opinion, this paper failed to adequately do this making it challenging to assess its contribution*. I discuss this in detail below.

**Contribution of this work**

My understanding of this paper's main contribution is that it leverages theoretical and empirical results in Kumar et. al, 2020, to directly assume that Property 1 (line 184) holds for VAEs. This property is then used to ultimately prove Lemma 5.1 and to then show that this result implies identifiability (Section 6).

Thus, as I understand, the main aspects of this work that should be understood as novel are Lemma 5.1/Theorem 5.2 for VAEs after assuming Property 1 holds, as well as the identifiability analysis based on these results in Section 6.

**Hessian Diagonality**

Firstly, to prove Lemma 5.1, the authors rely on the assumption of Property 1, which I understand to mean that the decoder's Jacobian is orthogonal and its Hessian diagonal? If this is the assumption, then it is important to note that there have been multiple works in the identifiability community which show identifiability for generators with a diagonal Hessian [10, 11]. Thus, it is unclear to me, at the moment, the extent to which this current identifiability result is novel relative to prior results given this Hessian assumption.

**Role of Beta**

Another stated contribution of the paper is understanding the role of the beta hyperparameter in VAEs for disentanglement as it relates to decoder variance (lines 48-49). As I understand, similar results were shown in Kumar et. al, 2020 [2] as well as Reizinger et. al, 2022 [5] (Appendix A.3). Thus, it is also not clear to me the extent to which this result is novel relative to prior works.

**Experiments**

An experimental study is conducted in Section 7 regarding the relationship between a VAEs derivative structure and disentanglement. My understanding, however, is that very similar experiments have been run in prior works such as [1, 2, 5]. I thus believe it is important for the authors to discuss better how their experiments differ from prior works.

**Disentanglement Definition**

In Section 3, disentanglement is defined to mean that the pushforward distribution of the prior under the decoder factorizes along component-wise pushforward distributions. This disentanglement definition seems tailored to the authors results and does not reflect definitions pursued in prior theoretical works [6] which ground disentanglement more rigorously in terms of identifiability. Thus, I believe more motivation for this definition and how it relates to prior works is needed.

**General framing of contribution**

In general, I believe the paper would benefit from presenting its contribution more precisely in the title, abstract, and introduction. Instead of using the broad phrasing/messaging of understanding disentanglement in VAEs, I believe clearly stating, from the outset, what the papers contribution is relative to prior work on VAEs and identifiability, and how it builds upon these works, would improve the paper.

**Conclusion**

For the reasons discussed above, I found it difficult to assess the contribution of this work relative to prior works. Thus, for the time being, I do not recommend acceptance.


**References**

[1] Variational Autoencoders Pursue PCA Directions (by Accident) (https://arxiv.org/abs/1812.06775)

[2] On Implicit Regularization in β-VAEs (https://arxiv.org/abs/2002.00041)

[3] Diagnosing and Enhancing VAE Models (https://arxiv.org/abs/1903.05789)

[4] Demystifying Inductive Biases for (Beta-)VAE Based Architectures (https://proceedings.mlr.press/v139/zietlow21a.html)

[5] Embrace the Gap: VAEs Perform Independent Mechanism Analysis (https://arxiv.org/abs/2206.02416)

[6] Nonlinear Independent Component Analysis for Principled Disentanglement in Unsupervised Deep Learning (https://arxiv.org/abs/2303.16535)

[7] Independent mechanism analysis, a new concept? (https://arxiv.org/abs/2106.05200)

[8] Function Classes for Identifiable Nonlinear Independent Component Analysis (https://arxiv.org/abs/2208.06406)

[9] Robustness of Nonlinear Representation Learning (https://arxiv.org/abs/2503.15355)

[10] Additive Decoders for Latent Variables Identification and Cartesian-Product Extrapolation (https://arxiv.org/abs/2307.02598)

[11] Interaction Asymmetry: A General Principle for Learning Composable Abstractions (https://arxiv.org/abs/2411.07784)

**Questions:**

* What do the authors view as their main contribution relative to prior works on VAEs and identifiability? Can the authors comment further on the relationship between their results and the prior identifiability and VAE results discussed above.


* Why are disentanglement and identifiability defined separately in the authors formalism. How do the authors justify this new definition of disentanglement?


* What novelty do the authors believe their experiments offer relative to prior works?

---

> ### Author Response · Authors · 2025-11-21
>
> We thank the reviewer for the constructive comments. We respond point‑wise below.
>
> ---
>
> **Summary of new vs prior work**: We:
>
> 1. give a *rigorous distributional definition of disentanglement* (Def. D1).
> 2. prove a *canonical factorization of the pushforward density* (Lem. 5.1) and *necessary‑and‑sufficient decoder conditions* for independence (Thm. 5.2).
> 3. prove *identifiabilty of  independent factors*, up to natural symmetries (§6).
> 4. show, by an exact identity, how *diagonal posteriors in VAEs promote those conditions* in expectation, and clarify the role of $\\beta$ (§4).
>
> Note, items **1–3 are general to a class of push-forwards, i.e. *VAE‑agnostic*** (no Property 1); item 4 links them to VAEs.
>
> ---
>
>
> **1) Main contribution vs prior VAEs/identifiability (Q1)**
>
> * **Clarification**: our core results (Def. D1; Lem. 5.1; Thm. 5.2; §6) **do not rely on [2] or assume Property P1**.
>    * P1 is brought in *after* Thm. 5.2 to connect these general results to Gaussian VAEs via an exact identity (§4), which supersedes the approximate result in [2].
> * **Positioning vs related VAE theory [1–5]:**
>    * prior works analyze Jacobian/Hessian statistics or inductive biases, and typically assume a Gaussian VAE to be unidentifiable based on ICA [6];
>    * **none provide** (i) a pushforward‑density factorization (Lem. 5.1), (ii) iff decoder conditions for independence (Thm. 5.2), or (iii) prove identifiability (§6).
> * **Positioning vs identifiability [6–11]:**
>    * [7–11] assume specific function classes (orthogonal Jacobians, additive/block‑Hessian decoders, interaction asymmetry).
>    * We instead *derive* decoder‑side conditions necessary and sufficient for independent features and *prove identifiability* under those conditions without assuming additivity/non‑overlap.
>
> *  We have **sharpened positioning of our work** in:
>    * the abstract, introductions, explicitly separating: (i) general pushforward/identifiability results (D1, Lem. 5.1, Thm. 5.2, §6) and (ii) their VAE instantiation via Property 1 and β (§4, §7).
>    * Related Works (§7) to more explicitly position our contributions as described above.
>
>
>
> ---
>
>
> **2) “Hessian diagonality” and relation to [10, 11]**
>
> * **Clarifications**:
>     * **P1 is not used in Lem. 5.1** (or Thm. 5.2).
>     * **C2 ≠ diagonal Hessian:** C2 states that each singular value (s^i(z)) depends only on its own latent coordinate $z_i$.
>         * This **does not** assume a diagonal Hessian in the ambient basis; rather, the **directed Hessian** (in the decoder’s tangent frame aligned with Jacobian singular vectors) is diagonalized (described in “Directed Hessian is tangent to the manifold”, App. A).
> * **Different assumptions to [10, 11]:** [10, 11] analyze **additive/compositional** decoders leading to block‑diagonal Hessians in pixel space, strictly **stronger constraints** than our Thm. 5.2.  Our conditions allow, e.g., rotations/scale/colour changes that would not be block‑diagonal in pixel space.
>
> * We have expanded Related Works to clarify.
>
> ---
>
> **3) Role of β; relation to [2, 5]**
>
> * That $\\beta$ modulates variance in a Gaussian VAE is well known and we do not claim novelty for it. Our contributions are
>    1. to **generalise the relationship to non-Gaussian VAEs** (App. B) linking to posterior collapse in *discrete* VAEs for language; and
>    1. to also show (via Price/Bonnet, §4), that **$\\beta$ rescales posteriors** and thereby the **latent space subject to disentanglement conditions C1–C2**.
>
>
> * We have updated the introduction and related works (§7) to clarify.

---

> ### Author Response · Authors · 2025-11-21
> **Author response (2/2)**
>
> **4) Experiments (Q3) and how they differ from [1, 2, 5]**
>
> *  Our paper is fundamentally theoretical and presents novel disentanglement results (outlined above). Given disentanglement has been well demonstrated, our **experiments serve to illustrate and target specific aspects of our theory**:
>
>   * in the (fully tractable) linear case (Fig. 2) we demonstrate **pushforward factorization, disentanglement and identifiability** of Lem. 5.1 / Thm. 5.2 / Thm 6.5, which apply equally to linear and non-linear VAEs;
>      * while linear identifiability is known from Lucas et al., we believe its demonstration is useful (a) to illustrate and (b) since it is not identifiable under standard ICA theory.
>   * on dSprites (known ground truth) we measure diagonality of the Price/Bonnet terms (Jacobian + **directed‑Hessian**), show how they vary with $\\beta$ and posterior structure, and correlate with disentangelement (Fig. 5);
>   * we have **added CelebA analysis** (App. F) to demonstrate disentanglement properties on a natural, more complex dataset.
>
>
> ---
>
>
>
> **5) Disentanglement definition (Q2) and separation from identifiability**
>
> * **Why D1:** Data are modelled as $x=g(z),\\ z\\sim \\prod_i p_i(z_i)$.
>    * A factorial prior does **not** imply observed independence because $g$ may entangle coordinates.
>    * D1 defines disentanglement as **factorization of the pushforward density** $p_{\\mu} = g_{\\\#} p$ into 1‑D seam factors, precisely the **distributional independence** one expects when “changing one factor leaves the others unaffected.”
> * **Why separate from identifiability:**
>   * **Disentanglement is a property of the learned distribution** (factorisation into component‑wise pushforwards);
>   * **identifiability is a property of the parameterization** (number of ways the property can be achieved).
>
>    Thm. 5.2 shows disentanglement (D1) $\\Leftrightarrow$ conditions C1–C2; §6 shows that under these conditions, factors are **identifiable** up to permuation/sign. This mirrors structure vs uniqueness in [6].
>
> * We have added the above description beneath defintion D1 to help the reader see how this definition fits naturally with statisitical independence and why that is relevant to disenatanglement.
>
> ---
>
> We hope these clarifications help the reviewer in assessing the novelty of our key contributions: **(i)** a distribution‑level definition of disentanglement (D1) based on canonical push-forward factorization (Lem. 5.1), **(ii)** *iff* decoder‑side conditions for disentanglement (Thm. 5.2), **(iii)** independent factor identifiability (Thm. 5.6), and **(iv)** an exact VAE mechanism via Price/Bonnet; none of which appear in prior VAE/identifiability work.

---

> > ### Comment · Reviewer_GVNd · 2025-11-27
> >
> > I appreciate the authors detailed reply and have raised my score to reflect this. I still have a few issues regarding the manuscript.
> >
> > &nbsp;
> >
> > * I did indeed misspeak as Lemma 5.1 and Theorem 5.2 don't rely on Properties 1 and 2. However, I think that my assessment of the contribution is still correct. Namely, the authors essentially start where Kumar and Poole [2] left off and show that this structure on the decoder Jacobian and Hessian implies some uniqueness of the learned decoder. Thus, I think the statement "we develop a full theoretical explanation of disentanglement and how it arises in VAEs" is a bit of an overclaim. As I understand, the authors show that the results from [2] imply a certain pushforward factorization which the authors *explicitly define as disentanglement*. They then show uniqueness of the decoder given this structure.
> >
> > &nbsp;
> >
> > * Regarding the relationship between Hessian diagonality and Property 1, can the authors please elaborate on the Hessian product in this assumption. I don't understand this current notation since the Hessian is a tensor.
> >
> > &nbsp;
> >
> > * Can the authors please elaborate on precisely what is novel in their experiments relative to prior works? Have all experiments (models and datasets) and quantities being measured been evaluated in prior works or is there something new being shown in these results
> >
> > &nbsp;
> >
> > * Reg. the claim "Disentanglement is a property of the learned distribution": What is this based on? This still feels tailored to the authors definition of disentanglement opposed to reflecting how the term is usually used in representation learning.

---

> ### Author Response · Authors · 2025-11-29
>
> We thank the reviewer for their acknowledgement and **raising their score**.
>
> * **Scope clarification**: The reviewer's understanding of the paper's logical flow is not quite correct, so we clarify (as now reflected in the abstract/intro):
>
>    1. **Disentanglement definition (D1) is VAE-agnostic and based on statistical independence**:
>       * if observations $x$ have observably independent generative factors $\\{g_i\\}$, then $p(x)$ is given by the product of their probabilities and so *factorises* (as in Eq 5),
>          e.g. $p(x=\\textit{big square}) \\;=\\; p(g_1\\!=\\!\\textit{big})\\,p(g_2\\!=\\!\\textit{square}).$
>       * For a pushforward $p_\\mu(x)$ used to model $p(x)$, we **define disentanglement as when distinct latent variables map to distinct generative factors** (e.g. $z_i$ maps to "size", $z_j$ to "shape" etc).
>       * D1 is therefore **grounded in statistical independence** and simply **formalises an intuition** alluded to in many disentanglement works (e.g. Higgins et al., 2017; Burgess et al., 2019; Chen et al., 2019).
>    2. While still **VAE-agnostic**, we prove *iff* conditions for disentanglement, C1–C2 (Thm. 5.2), and its identifiability (§6) for a general smooth push-forward;
>    3. **Only then do we recall that a VAE with diagonal posteriors induces C1-C2 and so disentanglement** (due to the exact Price/Bonnet identity, Eq 6, §4).
>   * Wording calibrated to “a **principled, distribution‑level** explanation of disentanglement ..." (replacing "full").
>
> * **On the relationship to Kumar & Poole [2] and scope of our claim**
>
>   * It can be readily veriied that **none of our results rely on [2]**: D1 is defined *from first principles* (as above); Lem. 5.1, Thm. 5.2, §6 are proved for *general smooth decoders*. The link to VAEs is via an exact identity (Eq 6).
>   * Prior VAE works, incl. [2], do not establish any of our core results.
>
>
> * **On the “Hessian product” in P1**
>
>    * Terms in Property P1 are from the identity in Eq 6 (known from, e.g. Opper and Archambeau, 2009).
>    * $\\mathbf{H}_z\\in\\mathbb{R}^{n\\times d\\times d}$ is the (coordinate-wise) decoder Hessian and $r\\!=\\!x\\!-\\!d(z)\\in\\mathbb{R}^n$, hence the contraction $r^\\top\\mathbf{H}_z \\!\\in\\mathbb{R}^{d\\times d}$ in Eq 6, or *directed Hessian*, is a square matrix.
>
>
> * **On experimental novelty relative to prior work**
>
>    * Our **main contribution is theoretic** (as above) to explain the many empirical observation of disentanglement in terms of the intrinsic structure of the data distribution.
>    * Since disentanglement has been widely observed, some aspects of our experiments are known and serve as a self-contained illustration of our theory. For example,
>       * having *proved* theoretically that latent variables seek to identify ground truth factors, despite associated *unidentifiability* proofs, we demonstrate that our claim holds in practice with heat maps of mutual information (Fig. 5).
>       * our clearer understanding of *why/how* $\\beta$ trades off output noise and disentanglement theoretically implies that annealing $\\beta$ (an empirical heuristic) should give lower-noise, disentangled samples, which we thus demonstrate.
>
>    * Aspects of the empirical support that are **novel relative to prior works** include:
>
>      * a full illustration of **independent component identifiability**, or breaking of Gaussian prior symmetry, due to diagonal covariances (Fig 2).
>         * This gives further clear support in a fully tractible setting for our proof of identifiability, even where classical/noisy‑ICA is *unidentifiable without auxiliary information*.
>      * we measure diagonality of *both* terms in the Price/Bonnet identity  under diagonal vs full posteriors (Fig 5), including the **directed Hessian** (further to known Jacobian orthogonality from Rolinek et al.; Kumar & Poole).
>
>
>
>
> * **Basis for “disentanglement is a property of the learned distribution”**
>
>    * As above, for data $x$ with independent generative factors, $p(x)$ factorises and D1 defines disentanglement as when a pushforward model (the "learned distribution") maps latent variables to those factors.
>    * For a pushforward $p_\\mu$ (defined by a factorial prior and a decoder) to match $p(x)$ and achieve disentanglement, $p_\\mu$ must factorise with distinct latent variables mapped to distinct factors.
>    * Thm 5.2 describes precisely the conditions on the decoder needed for that, hence disentanglement is a property of the learned model.
>
>
> We thank the reviewer again and hope this resolves any remaining concerns/misconceptions.
>
> We believe the fact that a Gaussian pushforward can be disentangled by a VAE, even when provably unidentifiable under classical/noisy ICA, is a novel, surprising result of broad interest given the many works on disentanglement/identifiability/ICA.

---

### Author Response · Authors · 2025-11-21
**Updated manuscript**

Dear all reviewers,

We have updated the manuscript to take account of reviewer comments (highlighted in teal). No changes have been made to the main content of the paper. The main changes can be summarised as:
* **generality**: highlighting more clearly that our main results apply more generally than to VAEs, i.e. to smooth pushforwards (title, abstract, intro, conclusion)
* **motivation for definition D1** (disentanglement)
* **positioning** relative to other works (Related Works)
* **empirical analysis** included for CelebA

We respond to each reviewer in detail below and look forward to further feedback.

---

### Author Response · Authors · 2025-12-03
**Note to AC/SAC given unusual circumstances**

Dear AC/SAC,

Many thanks for reviewing our work. Given the unusual circumstances and increased AC burden, we give a brief overview of the paper and reviews:

---

**Paper summary**: our work gives a **principled, distribution‑level explanation of disentanglement** in terms of *identifiable latent factors*, and explains how it arises in VAEs. We:
1. introduce a **rigorous definition of disentanglement** (D1) as factorisation of the data density into independent 1-D “seams”;
2. prove *iff* decoder conditions (C1/C2) that say each latent coordinate controls its own factor independent of others; and
3. show that **underlying factors are *identifiable*** (up to symmetry) even with a Gaussian prior, addressing long‑standing unidentifiability concerns (from ICA).

These are all *general to push-forward distributions* (e.g. as also in GANs); we then show how (ß-)VAEs with diagonal posteriors induce C1/C2 and so disentanglement (controlled by β).

Since disentanglement has been widely demonstrated, our experiments are targeted to our theory. In controlled settings (Gaussian, dSprites) and now CelebA, we show diagonal posteriors break Gaussian prior symmetry to align latent axes with underlying factors (i.e. *disentangle*) and under-the-hood promote diagonality of derivative terms (our *iff* conditions), while maintaining comparable quality to full‑covariance VAEs.

---

**Reviewer GVNd**
   * [+] find the main theory **"clear and well structured"** and regard our main lemmas/identifiability results as "potentially interesting"
   * [-] was initially unclear on a few aspects, e.g. what is new relative to prior VAE/identifiability work, our disentanglement definition, the Hessian term, experiment novelty.
       * These largely stemmed from misunderstanding, addressed through rebuttal and updated paper. The reviewer acknowledged these clarifications (agreeing they "misspoke") and increased their score.

**Reviewer iHbB**:
   * [+] gives a positive conceptual summary and values the precise disentanglement definition and link to diagonal posteriors.
   * [-] believed our proofs were "limited to the linear case" and that we repeated known observations of ß.
      * These largely reflect misunderstandings, addressed through rebuttal (our core results are all *non‑linear and model‑agnostic*; β is discussed in terms of our novel disentanglement definition; our theory complements rather than competes with methods like β‑TCVAE) and updated paper.

**Reviewer GTYe**:
   * [+] evaluates the paper positively, notes **novelty of proving identifiability** and of tying disentanglement in VAEs to diagonal posteriors and β.
   * [-] main concern: experiments limited to simple settings (linear‑Gaussian, dSprites, small VAEs), not more complex natural images.
      * We frame the paper as primarily theoretical (similar to and advancing a line of published research); clarify the empirical support we give, beyond the many known empirical findings we explain; and address the main concern by adding a natural data set (CelebA) to illustrate disentanglement/aspects of our theory.

**Reviewer 8oT4**:
   * [+] evaluates the paper positively, views theoretical results as **a solid foundation for understanding disentanglement** and values the correction of prior work
   * [-] considers experimental support relatively light and asked for more insight (e.g. latent dimension for full disentanglement, relation to noisy ICA and independent factors, impact of C1/C2 on reconstruction/generation).
       * We expand discussion to include latent dimensionality and noisy ICA; add experiments on natural data (CelebA) to address the core concern; and clarify points (diagonal posteriors select among equivalent optima hence reconstruction/generation quality for diagonal/full‑covariance VAEs are comparable; active dimensions), while maintaining the paper's theoretical focus.

We believe we have addressed all concerns raised and clarified how our work advances understanding of disentanglement, as a distributional concept, and its identifiability (of particular interest given ICA *unidentifiability*).

Best,
Authors

---

### Meta-Review · Area_Chair_5AUx · 2025-12-25

**Summary:**

The paper defines a notion of disentanglement in VAEs, and shows sufficient conditions on the decoder for it to obtain. It further analyzes the special case of linear (Gaussian) VAEs with diagonal posteriors and beta-regularization, and makes a case for why disentanglement might naturally arise in practice.

The discussion with the reviewers, which I agree with, is that the paper is not ready for publication. Several reviewers (iHbB, GTYe, 8oT4) point out that the paper's implication for improving training VAE training "in the wild", and more generally, the practice of VAEs is unclear. The authors conceded this point, and suggested that their intent is to primarily provide a theoretical foundation for talking about disentanglement. But in this case, the discussion with reviewer GVNd makes it clear that the paper needs to more clearly position itself in relation to prior work (of which there is quite a lot). The authors added some clarifications in the discussion and the edited paper --- but I believe there are still outstanding points that are unresolved, and the paper is not ready.

**Reviewer Concerns:**

Impact to empirics, positioning to prior theory works on disentangling.

**Reviewer Scores:**

The points of contention (positioning with respect to prior theory, empirical impact) weren't really resolved in the discussion, so I doubt reviewers would have improved their score.

---

### Decision · Program_Chairs · 2026-01-26

Reject